# An Improved Grey Wolf Optimizer and Its Application in Robot Path Planning

**DOI:** 10.3390/biomimetics8010084

**Published:** 2023-02-16

**Authors:** Yun Ou, Pengfei Yin, Liping Mo

**Affiliations:** 1School of Communication and Electronic Engineering, Jishou University, Jishou 416000, China; 2Academic Affairs Office, Jishou University, Jishou 416000, China; 3College of Computer Science and Engineering, Jishou University, Jishou 416000, China

**Keywords:** grey wolf optimizer, clonal selection algorithm, position-updating strategy, nonlinear function, robot path planning

## Abstract

This paper discusses a hybrid grey wolf optimizer utilizing a clone selection algorithm (pGWO-CSA) to overcome the disadvantages of a standard grey wolf optimizer (GWO), such as slow convergence speed, low accuracy in the single-peak function, and easily falling into local optimum in the multi-peak function and complex problems. The modifications of the proposed pGWO-CSA could be classified into the following three aspects. Firstly, a nonlinear function is used instead of a linear function for adjusting the iterative attenuation of the convergence factor to balance exploitation and exploration automatically. Then, an optimal *α* wolf is designed which will not be affected by the wolves *β* and *δ* with poor fitness in the position updating strategy; the second-best *β* wolf is designed, which will be affected by the low fitness value of the *δ* wolf. Finally, the cloning and super-mutation of the clonal selection algorithm (CSA) are introduced into GWO to enhance the ability to jump out of the local optimum. In the experimental part, 15 benchmark functions are selected to perform the function optimization tasks to reveal the performance of pGWO-CSA further. Due to the statistical analysis of the obtained experimental data, the pGWO-CSA is superior to these classical swarm intelligence algorithms, GWO, and related variants. Furthermore, in order to verify the applicability of the algorithm, it was applied to the robot path-planning problem and obtained excellent results.

## 1. Introduction

The metaheuristic algorithm is an improvement of the heuristic algorithm combined with a random algorithm and local search algorithm to implement optimization tasks. In recent years, metaheuristic optimization has made some recent developments. Jiang X et al. proposed optimal pathfinding with a beetle antennae search algorithm using ant colony optimization initialization and different searching strategies [1]. Khan A H et al. proposed BAS-ADAM: an ADAM-based approach to improving the performance of beetle antennae search optimizer [2]. Ye et al. proposed a modified multi-objective cuckoo search mechanism and applied this algorithm to the obstacle avoidance problem of multiple uncrewed aerial vehicles (UAVs) for seeking a safe route by optimizing the coordinated formation control of UAVs to ensure the horizontal airspeed, yaw angle, altitude, and altitude rate are converged to the expected level within a given time for inverse kinematics and optimization [3]. Khan et al. proposed using the social behavior of beetles to establish a computational model for operational management [4]. As one of the latest metaheuristic algorithms, grey wolf optimizer (GWO) is widely employed to settle real industrial issues because GWO maintains a balance between exploitation and exploration through dynamic parameters and has a strong ability to explore the rugged search space of the problem [5,6], such as the selection problem [7,8,9], privacy protection issue [10], adaptive weight problem [11], smart home scheduling problem [12], prediction problem [13,14,15], classification problem [16], and optimization problem [17,18,19,20].

Although the theoretical analysis and industrial applications using GWO have gained fruitful achievements, there still exist some shortcomings that hinder the further developments of GWO, such as slow convergence speed and low accuracy in the single-peak function and easily falling into local optimum in the multi-peak function and complex problems. Recently, various GWO variants have been investigated to overcome the mentioned shortages. Mittal et al. proposed a modified GWO (mGWO) to solve the balance problem between the exploitation and exploration of GWO. The main contribution is to propose an exponential function instead of a linear function to adjust the iterative attenuation of parameter *a*. Experimental results proved that the mGWO improves the effectiveness, efficiency, and stability of GWO [21]. Saxena et al. proposed a β-GWO to improve the exploitation and exploration of GWO by embedding a β-chaotic sequence into parameter *a* through a normalized mapping method. Experimental results demonstrated that β-GWO had good exploitation and exploration performance [22]. Long et al. proposed the exploration-enhanced GWO (EEGWO) to overcome GWO’s weakness of good exploitation but poor exploration based on two modifications. Meanwhile, a random individual is used to guide the search for new individual candidates. Furthermore, a nonlinear control parameter strategy is introduced to obtain a good exploitation effect and avoid poor exploration effects. Experimental results illustrated that the proposed EEGWO algorithm significantly improves the performance of GWO [23]. Gupta and Deep proposed an RW-GWO to improve the search capability of GWO. The main contribution is to propose an improved method based on a random walk. Experimental results showed that RW-GWO provides a better lead in searching for grey wolf prey [24]. Teng et al. proposed a PSO_GWO to solve the problem of slow convergence speed and low accuracy of the grey wolf optimization algorithm. The main contribution can be divided into three aspects: firstly, a Tent chaotic sequence is used to initialize individual positions; secondly, a nonlinear control parameter is used; finally, particle swarm optimization (PSO) is combined with GWO. Experimental results showed that PSO_GWO could better search the optimal global solution and have better robustness [25]. Gupta et al. proposed SC-GWO to solve the balance problem of exploitation and exploration. The main contribution is the combination of the sine and cosine algorithm (SCA) and GWO. Experimental results showed that SC-GWO has good robustness to problem scalability [26]. In order to improve the performance of GWO in solving complex and multimodal functions, Yu et al. proposed an object-based learning wolf optimizer (OGWO). Without increasing the computational complexity, the algorithm integrates the opposing learning method into GWO in the form of a jump rate, which helps the algorithm jump out of the local optimum [27]. To improve the iterative convergence speed of GWO, Zhang et al. also improved the algorithm flow of GWO and proposed two dynamic GWOs (DGWO1 and DGWO2) [28].

Although these GWO variants improve the convergence speed and accuracy in the single-peak function and have the ability to jump out of local optimum in the multi-peak function and complex problems, they still have the disadvantages of slow convergence speed and low accuracy and easily falling into local optimum while solving some complex problems. To overcome these shortages, an improved GWO is proposed in this paper by combining it with a clonal selection algorithm (CSA) to improve the convergence speed, accuracy, and jump out of the local optimum of standard GWO. The proposed algorithm is called pGWO-CSA. The core improvements could be classified into the following points:

Firstly, a nonlinear function is used instead of a linear function for adjusting the iterative attenuation of the convergence factor to balance exploitation and exploration automatically.

Secondly, the pGWO-CSA adopts a new position updating strategy, and the position updating of *α* wolf is no longer affected by the wolves *β* and *δ* with poor fitness. The position updating of the *β* wolf is no longer affected by the low fitness value of the *δ* wolf.

Finally, the pGWO-CSA combines GWO with CSA and introduces the cloning and super-mutation of the CSA into GWO to improve GWO’s ability to jump out of local optimum.

In the experimental part, 15 benchmark functions are selected to perform the function optimization tasks to reveal the performance of pGWO-CSA further. Firstly, pGWO-CSA is compared with other swarm intelligence algorithms, particle swarm optimization (PSO) [29], differential evolution (DE) [30], and firefly algorithm (FA) [31]. Then pGWO-CSA is compared with GWO [5] and its variants OGWO [27], DGWO1, and DGWO2 [28]. Due to the statistical analysis of the obtained experimental data, the pGWO-CSA is superior to these classical swarm intelligence algorithms, GWO, and related variants.

The rest sections are organized as follows. Section 2 introduces the GWO and CSA, Section 3 introduces the improvement ideas and reasons for pGWO-CSA in detail, Section 4 mainly introduces experimental tests, Section 5 introduces the robot path-planning problem, and Section 6 is the summary of the whole paper.

## 2. GWO and CSA

This section mainly introduces the relevant concepts and algorithm ideas of GWO and CSA to provide theoretical support for subsequent improvement research.

### 2.1. GWO

GWO is a swarm intelligence algorithm proposed by Mirjalili et al. in 2014, which is inspired by the hunting behavior of grey wolves [5]. In nature, grey wolves like to live in packs and have a very strict social hierarchy. There are four types of wolves in the pack, ranked from highest to lowest in the social hierarchy: the *α* wolf, *β* wolf, *δ* wolf, and ω wolf. The GWO is also based on the social hierarchy of grey wolves and their hunting behavior, and its specific mathematical model is as follows.

(1)Surround the prey

In the process of hunting, in order to surround the prey, it is necessary to calculate the distance between the current grey wolf and the prey and then update the position according to the distance. The behavior of grey wolves rounding up prey is defined as follows:(1)$$X(t+1)=X_{P}(t)-A\times D$$
and
(2)$$D=|C\times X_{P}(t)-X(t)|,$$
where Formula (1) is the updating formula of the grey wolf’s position, and Formula (2) is the calculation formula of the distance between the grey wolf individual and prey. Variable *t* is the current iteration number, $X_{P}(t)$ and X(t) are the current position vectors of the prey and the grey wolf at iteration *t*, respectively. A and C are coefficient vectors calculated by Formula (3) and Formula (4), respectively.
(3)$$A=2\times a\times r_{1}-a,$$
(4)$$C=2\times r_{2},$$
and
(5)$$a=2-2\times \frac{t}{t_{\max}},$$
where a is the convergence factor, and a linearly decreases from 2 to 0 as the number of iterations increases. $r_{1}$ and $r_{2}$ are random vectors in [0, 1]. Formula (5) is the calculation formula a and $t_{\max}$ indicates the maximum number of iterations.

(2)Hunting

In an abstract search space, the position of the optimal solution is uncertain. In order to simulate the hunting behavior of grey wolves, *α*, *β*, and *δ* wolves are assumed to have a better understanding of the potential location of prey. *α* wolf is regarded as the optimal solution, *β* wolf is regarded as the suboptimal solution, and *δ* wolf is regarded as the third optimal solution. Other gray wolves update their positions based on *α*, *β*, and *δ* wolves, and the calculation formulas are as follows:(6)$$\begin{array}{c} D_{\alpha }=|C_{1}\times X_{\alpha }-X(t)|\\ D_{\beta }=|C_{2}\times X_{\beta }-X(t)|\\ D_{\delta }=|C_{3}\times X_{\delta }-X(t)| \end{array},$$
(7)$$\begin{array}{c} X_{1}=X_{\alpha }-A_{1}\times D_{\alpha }\\ X_{2}=X_{\beta }-A_{2}\times D_{\beta }\\ X_{3}=X_{\delta }-A_{3}\times D_{\delta } \end{array},$$
and
(8)$$X(t+1)=(X_{1}+X_{2}+X_{3})/3,$$
where $D_{\alpha }$ represents the distance between the current grey wolf and *α* wolf; $D_{\beta }$ represents the distance between the current grey wolf and *β* wolf; $D_{\delta }$ represents the distance between the current grey wolf and *δ* wolf; and $X_{\alpha }$, $X_{\beta }$, and $X_{\delta }$ represent the position vectors of *α* wolf, *β* wolf, and *δ* wolf, respectively. X(t) is the current position of the grey wolf. $C_{1}$, $C_{2}$, and $C_{3}$ are random vectors, calculated by Formula (4). $A_{1}$, $A_{2}$, and $A_{3}$ are determined by Formula (3). Formula (7) represents the step length and direction of grey wolf individuals to *α*, *β*, and *δ* wolves, and Formula (8) is the position-updating formula of grey wolf individuals.

According to the description above, the algorithm flow chart of GWO is shown in Figure 1.

### 2.2. CSA

The CSA was proposed by De Castro and Von Zuben in 2002 according to the clonal selection theory [32]. The CSA simulates the mechanism of immunological multiplication, mutation, and selection and is widely used in many problems.

For the convenience of model design, the principle of the biological immune system is simplified. All substances that do not belong to themselves are regarded as antigens. When the immune system is stimulated by antigens, antibodies will be produced to bind to antigens specifically. The stronger the association between antigen and antibody, the higher the affinity. Then, the antibodies with high antigen affinity are selected to multiply and differentiate between binding to the antigens, increase their antigen affinity through super-mutation, and finally eliminate the antigens. In addition, some of the antibodies are converted into memory cells in order to respond quickly to the same or similar antigens in the future. In the CSA, the problem that needs to be solved is regarded as the antigen, and the solution to the problem is regarded as the antibody. At the same time, the receptor-editing mechanism is adopted to avoid falling into the local optimum. The flow chart of CSA is shown in Figure 2.

The above is the introduction of the GWO and CSA. The proposed algorithm in this paper is also inspired by GWO and CSA. The pGWO-CSA proposed in this paper is introduced in detail in Section 3 below.

## 3. The Proposed pGWO-CSA

In order to improve the convergence speed and accuracy in the single-peak function and the ability to jump out of local optimum in the multi-peak function and complex problems: Firstly, a nonlinear function is used instead of a linear function for adjusting the iterative attenuation of convergence factor to balance exploitation and exploration automatically; Secondly, the pGWO-CSA adopts a new position-updating strategy, and different position-updating strategies are used for *α* wolf, *β* wolf, and other wolves, so that the position updating of *α* wolf and *β* wolf are not affected by the wolves with lower fitness; Finally, the pGWO-CSA combines GWO with CSA and introduce the cloning and super-mutation of the CSA into GWO.

The detailed improvement strategy is as follows.

### 3.1. Replace Linear Function with Nonlinear Function

In GWO, a decreases from 2 to 0 as the number of iterations increases, and the range of A decreases as a decreases. According to Formulas (6) and (7), when |A|<1, the next position of the grey wolf can be anywhere between the current position and the prey, and the grey wolf approaches the prey guided by *α*, *β*, and *δ*. When |A|≥1, the grey wolf moves away from the current *α*, *β*, and *δ* wolves and searches for the optimal global value. Therefore, when |A|<1, grey wolves approach their prey for exploitation. When |A|≥1, grey wolves move away from their prey for exploration. In the original GWO, the parameter a linearly decreases from 2 to 0, with half of the iterations devoted to exploitation and half to exploration. In order to balance exploitation and exploration, the pGWO-CSA adopts a nonlinear function instead of a linear function to adjust the iterative attenuation of parameter a so as to enhance the exploration ability of the grey wolf at the early stage of iteration. In pGWO-CSA, parameter a is calculated by Formula (9).
(9)$$a=\cos\left(\pi \times {\left(\frac{t}{t_{\max}}\right)}^{u}\right)+1.0,$$
where variable *t* is the current iteration number, $t_{\max}$ is the maximum iteration number, and *u* is the coefficient, where the value in this paper is 2.

Iterative curves of parameter a in the original GWO and pGWO-CSA are shown in Figure 3.

As can be seen from Figure 3, the convergence factor slowly decays in the early stage, improving the global search ability, and rapidly decays in the later stage, accelerating the search speed and optimizing the global exploration and local development ability of the algorithm.

### 3.2. Improve the Grey Wolf Position Updating Strategy

In the original grey wolf algorithm, the positions of all grey wolves in each iteration are updated by Formulas (6)–(8). In the position-updating strategy, the position updating of the *α* wolf is affected by the *β* wolf and *δ* wolf with poor fitness. The position updating of the *β* wolf is affected by the *δ* wolf with poor fitness.

Therefore, a new location updating strategy is proposed in this paper. In each iteration, the fitness of grey wolves is calculated, and the top three wolves *α*, *β*, and *δ* with the best fitness are saved and recorded. The specific location update formula is as follows.
(10)$$X^{\prime }(t)=\{\begin{array}{ccc} X_{1} & & \mathrm{if}\;\alpha \;\mathrm{wolf}\\ (X_{1}+X_{2})/2 & & \mathrm{if}\;\beta \;\mathrm{wolf}\\ (X_{1}+X_{2}+X_{3})/3 & & \mathrm{otherwise} \end{array},$$
where $X_{1}$, $X_{2}$, and $X_{3}$ are determined by Formula (7). $X^{\prime }(t)$ represents the pre-update position. On this basis, if the current *α* wolf and *β* wolf are close to the optimal solution, *α* wolf and *β* wolf have a greater probability to update to the position with better fitness so as to better guide wolves to hunt the prey and find the optimal solution. If *α* wolf and *β* wolf are in the local optimum, other wolves still update their positions according to Formulas (6)–(8) so that algorithm will not fall into the local optimum. Therefore, the proposed improved method can not only improve the exploitation capability but also not affect the exploration capability.

### 3.3. Combine GWO with CSA

GWO is combined with CSA by introducing the cloning and super-mutation of the CSA into GWO, and the exploitation and exploration ability of GWO is improved. For each grey wolf, a super-mutation coefficient (*Sc*) and a random number (*r*3) are introduced. The wolf with good adaptability has a small coefficient of super-variance and a small probability of variation, while the wolf with poor adaptability has a large probability of variation. If the super-variation coefficient Sc of the current grey wolf is greater than the random number *r*3, the current grey wolf will be cloned, and then the cloned grey wolf will be mutated through Formulas (6)–(8). If the mutated grey wolf has comparatively better adaptability, it will replace the current grey wolf. The specific calculation formula is as follows.
(11)$$Sc=\frac{Fitness_{i}-Fitness_{\min}}{Fitness_{\max}-Fitness_{\min}}+0.1$$
and
(12)$$X(t+1)=\{\begin{array}{c} X^{\prime }(t)\begin{array}{cc} , & \mathrm{if}\;Sc\leq r3 \end{array}\\ X^{\prime \prime }(t)\begin{array}{cc} , & \mathrm{otherwise} \end{array} \end{array},$$
where $fitness_{i}$ represents the fitness of the current grey wolf, $fitness_{\min}$ represents the fitness of the best wolf, and $fitness_{\max}$ represents the fitness of the worst wolf. r3 is a random number between [0, 1]. $X^{\prime }(t)$ is determined by Formula (10). $X^{\prime \prime }(t)$ represents the best between $X^{\prime }(t)$ and $X^{\prime }(t)$ as the result of the variation of X(t) through Formulas (6)–(8).

### 3.4. Algorithm Flow Chart of pGWO-CSA

According to the improvement idea mentioned above, the algorithm flow chart of pGWO-CSA is shown in Figure 4.

### 3.5. Time Complexity Analysis of the Algorithm

Assuming that the population size is N, the dimension of objective function F is Dim, and the number of iterations is T, the time complexity of the pGWO-CSA algorithm can be calculated as follows.

First, the time complexity required to initialize the grey wolf population is O(N × Dim), the time complexity required to calculate the fitness of all grey wolves is O(N × F (Dim)), and the time complexity required to preserve the location of the best three wolves is O(3 × Dim).

Then, in each iteration, the time complexity required to complete all grey wolves’ position updating is O(N × Dim), the time complexity required to update a, A, and C is O(1), and the time complexity required to calculate the fitness of all grey wolves is O(N × F(Dim)). The time complexity of cloning and super-mutation is O(N1 × Dim), where N1 is the number of wolves meeting the mutating condition, and the time complexity of updating the fitness and location of *α*, *β*, and *δ* is O(3 × Dim). The total iteration is T times. So, the total time complexity is O(T × N × Dim) + O(T) + O(T × N × F (Dim)) + O(T × N × Dim) + O(T × 3 × Dim).

So, in the worst case, the time complexity of the whole algorithm is O(N × Dim) + O(N × F(Dim)) + O(3 × Dim) + O(T × N × Dim) + O(T) + O(T × N × F(Dim)) + O(T × N × Dim) + O(T × 3 × Dim)≈O(T × N × (Dim + F(Dim))).

## 4. Experimental Test

In this section, 15 benchmark functions from F1–F15 are selected to test the performance of pGWO-CSA. Firstly, pGWO-CSA is compared with other swarm intelligence algorithms. Then pGWO-CSA is compared with GWO and its variants. Table 1 describes these benchmark functions in detail. Section 4.1 will compare pGWO-CSA with other swarm intelligence algorithms. Section 4.2 will compare pGWO-CSA with GWO and its variants.

Among these benchmark functions, the first seven benchmark functions, F1–F7, are simpler, while the last eight benchmark functions, F8–F15, are more complex. The dimension of these benchmark functions is 30 dimensions and 50 dimensions. The population size of all algorithms is set to 30, the maximal iteration of all algorithms is set to 500, and all experimental data are measured on the same computer to ensure a fair comparison between different algorithms. In order to avoid the randomness of the algorithm, each algorithm in this paper will be run on each test function 30 times. At the same time, the mean, standard deviation, and minimum and maximum values of the running results are recorded.

### 4.1. Compare with Other Swarm Intelligence Algorithms

In the comparison between pGWO-CSA and other swarm intelligence algorithms, PSO [29], DE [30], and FA [31] are selected to compare with pGWO-CSA. For each algorithm, the function optimization task is performed on the test functions F1–F15 in 30 dimensions and 50 dimensions, and the mean, standard deviation, and minimum and maximum values of the running results are recorded. The main parameters of the PSO, DE, FA, and pGWO-CSA are shown in Table 2.

In 30 dimensions, the test results of these four algorithms on test function F1–F15 are shown in Table 3. The convergence curves of these four algorithms on test function F1–F15 are recorded in Figure 5a, Figure 6a, Figure 7a, Figure 8a, Figure 9a, Figure 10a, Figure 11a, Figure 12a, Figure 13a, Figure 14a, Figure 15a, Figure 16a, Figure 17a, Figure 18a and Figure 19a.

In terms of the performance of the mean. Sort by the number of optimal values. The pGWO-CSA ranked first with 13 optimal values. PSO and DE tied for second place with one optimal value. FA ranked fourth with zero optimal values. Compared with PSO, pGWO-CSA outperformed PSO in 14 out of 15 test functions, and PSO outperformed pGWO-CSA only in test function F14. Compared with DE, pGWO-CSA outperformed DE in 14 out of 15 test functions, and DE outperformed pGWO-CSA only in test function F6. Compared with FA, pGWO-CSA outperformed FA in all 15 test functions.

In terms of the performance of the standard deviation. Sort by the number of optimal values. The pGWO-CSA ranked first with 11 optimal values. PSO ranked second with three optimal values. FA ranked third with one optimal value. DE ranked fourth with zero optimal values. Compared with PSO, pGWO-CSA outperformed PSO in 11 out of 15 test functions, and PSO outperformed pGWO-CSA in test functions F5, F6, F8, and F14. Compared with DE, pGWO-CSA outperformed DE in 13 out of 15 test functions, and DE outperformed pGWO-CSA in test functions F6 and F8. Compared with FA, pGWO-CSA outperformed FA in 14 out of 15 test functions, and FA outperformed pGWO-CSA only in test function F6.

In terms of the performance of the minimum. Sort by the number of optimal values. The pGWO-CSA ranked first with 13 optimal values. PSO ranked second with four optimal values. DE and FA tied for third place with zero optimal values. Compared with PSO, pGWO-CSA outperformed PSO in 11 out of 15 test functions, and PSO outperformed pGWO-CSA in test functions F7 and F14. In addition, the minimum values of pGWO-CSA and PSO are the same on the test functions F9 and F12, and both pGWO-CSA and PSO find the minimum values of the functions. Compared with DE and FA, pGWO-CSA outperformed DE and FA in all 15 test functions.

In terms of the performance of the maximum. Sort by the number of optimal values. The pGWO-CSA ranked first with 12 optimal values. DE ranked second with two optimal values. PSO ranked third with one optimal value. FA ranked fourth with zero optimal values. Compared with PSO, pGWO-CSA outperformed PSO in 14 out of 15 test functions, and PSO only outperformed pGWO-CSA in test function F14. Compared with DE, pGWO-CSA outperformed DE in 13 out of 15 test functions, and DE outperformed pGWO-CSA in test functions F6 and F8. Compared with FA, pGWO-CSA outperformed FA in 14 out of 15 test functions, and FA outperformed pGWO-CSA only in test function F6.

In addition, the pGWO-CSA can find theoretical optimal values on the test functions F9, F11, and F12. In the test functions F1, F2, F3, F4, F10, F11, F13, and F15, pGWO-CSA is superior to PSO, DE, and FA in terms of the mean, standard deviation, and minimum and maximum. Although PSO outperformed pGWO-CSA in the mean, standard deviation, and minimum and maximum on test function F14, pGWO-CSA still outperformed DE and FA on test function F14.

In 50 dimensions, the test results of these four algorithms on test function F1–F15 are shown in Table 4. The convergence curves of these four algorithms on test function F1–F15 are recorded in Figure 5b, Figure 6b, Figure 7b, Figure 8b, Figure 9b, Figure 10b, Figure 11b, Figure 12b, Figure 13b, Figure 14b, Figure 15b, Figure 16b, Figure 17b, Figure 18b and Figure 19b.

In terms of the performance of the mean. Sort by the number of optimal values. The pGWO-CSA ranked first with 13 optimal values. PSO and FA tied for second place with one optimal value. DE ranked fourth with zero optimal values. Compared with PSO, pGWO-CSA outperformed PSO in 14 out of 15 test functions, and PSO outperformed pGWO-CSA only in test function F14. Compared with DE, pGWO-CSA outperformed DE in all 15 test functions. Compared with FA, pGWO-CSA outperformed FA in 14 out of 15 test functions, and FA outperformed pGWO-CSA only in test function F6.

In terms of the performance of the standard deviation. Sort by the number of optimal values. The pGWO-CSA ranked first with 11 optimal values. PSO ranked second with three optimal values. FA ranked third with one optimal value. DE ranked fourth with zero optimal values. Compared with PSO, pGWO-CSA outperformed PSO in 11 out of 15 test functions, and PSO outperformed pGWO-CSA in test functions F5, F6, F8, and F14. Compared with DE, pGWO-CSA outperformed DE in 14 out of 15 test functions, and DE outperformed pGWO-CSA only in test function F8. Compared with FA, pGWO-CSA outperformed FA in 14 out of 15 test functions, and FA outperformed pGWO-CSA only in test function F6.

In terms of the performance of the minimum. Sort by the number of optimal values. The pGWO-CSA ranked first with 11 optimal values. PSO ranked second with four optimal values. FA ranked third with two optimal values. DE ranked fourth with zero optimal values. Compared with PSO, pGWO-CSA outperformed PSO in 11 out of 15 test functions, and PSO outperformed pGWO-CSA in test functions F7 and F14. In addition, the minimum value of pGWO-CSA and PSO are the same on the test functions F9 and F12, and both pGWO-CSA and PSO find the minimum values of the functions. Compared with DE, pGWO-CSA outperformed DE in all 15 test functions. Compared with FA, pGWO-CSA outperformed FA in 13 out of 15 test functions, and FA outperformed pGWO-CSA in test functions F6 and F8.

In terms of the performance of the maximum. Sort by the number of optimal values. The pGWO-CSA ranked first with 12 optimal values. PSO, DE, and FA are tied for second place with one optimal value. Compared with PSO, pGWO-CSA outperformed PSO in 14 out of 15 test functions, and PSO only outperformed pGWO-CSA in test function F14. Compared with DE, pGWO-CSA outperformed DE in 14 out of 15 test functions, and DE only outperformed pGWO-CSA in test function F8. Compared with FA, pGWO-CSA outperformed FA in 14 out of 15 test functions, and FA outperformed pGWO-CSA only in test function F6.

In addition, pGWO-CSA can find theoretical optimal values on the test functions F9, F11, and F12. In the test functions F1, F2, F3, F4, F9, F10, F13, and F15, pGWO-CSA is superior to PSO, DE, and FA in terms of the mean, standard deviation, and minimum and maximum. Although pGWO-CSA is not as good as FA on test function F6 and as good as PSO on test function F14, pGWO-CSA still outperformed the other two swarm intelligence algorithms on these two functions.

Based on the above data and analysis, pGWO-CSA has faster convergence speed, higher accuracy, and better ability to jump out of local optimum compared with other swarm intelligence algorithms in either 30 or 50 dimensions. In order to further verify the performance of pGWO-CSA, we will next compare pGWO-CSA with GWO and its variants.

### 4.2. Compare with GWO and Its Variants

In order to further verify the performance of pGWO-CSA, pGWO-CSA is compared with GWO [5] and its variants OGWO [27], DGWO1, and DGWO2 [28] on the test functions F1–F15 in 30 dimensions and 50 dimensions. The main parameters of pGWO-CSA, GWO, OGWO, DGWO1, and DGWO2 are shown in Table 5.

In 30 dimensions, the test results of these five algorithms on test function F1–F15 are shown in Table 6, with the optimal values highlighted in bold. The convergence curves of these five algorithms on test function F1–F15 are recorded in Figure 20a, Figure 21a, Figure 22a, Figure 23a, Figure 24a, Figure 25a, Figure 26a, Figure 27a, Figure 28a, Figure 29a, Figure 30a, Figure 31a, Figure 32a, Figure 33a and Figure 34a.

In 50 dimensions, the test results of these five algorithms on test function F1–F15 are shown in Table 7. The convergence curves of these five algorithms on test function F1–F15 are recorded in Figure 20b, Figure 21b, Figure 22b, Figure 23b, Figure 24b, Figure 25b, Figure 26b, Figure 27b, Figure 28b, Figure 29b, Figure 30b, Figure 31b, Figure 32b, Figure 33b and Figure 34b.

In terms of the performance of the mean. Sort by the number of optimal values. The pGWO-CSA ranked first with nine optimal values. DGWO2 ranked second with six optimal values. OGWO ranked third with two optimal values. DGWO1 and GWO tied for fourth place with one optimal value. Compared with GWO, pGWO-CSA outperformed GWO in 14 out of 15 test functions. Compared with OGWO, pGWO-CSA outperformed OGWO in 13 out of 15 test functions, and OGWO outperformed pGWO-CSA only in test function F7. Compared with DGWO1, pGWO-CSA outperformed DGWO1 in 14 out of 15 test functions. Compared with DGWO2, pGWO-CSA outperformed DGWO2 in 9 out of 15 test functions, and DGWO2 outperformed pGWO-CSA only in test functions F1, F2, F3, F4, and F14.

In terms of the performance of the standard deviation. Sort by the number of optimal values. The pGWO-CSA and DGWO2 tied for first place with seven optimal values. DGWO1 and OGWO tied for third place with two optimal values. GWO ranked fifth with one optimal value. Compared with GWO, pGWO-CSA outperformed GWO in 11 out of 15 test functions, and GWO only outperformed pGWO-CSA in test functions F7 and F15. Compared with OGWO, pGWO-CSA outperformed OGWO in 13 out of 15 test functions, and OGWO only outperformed pGWO-CSA in test function F7. Compared with DGWO1, pGWO-CSA outperformed DGWO1 in 13 out of 15 test functions, and DGWO1 only outperformed pGWO-CSA in test function F15. Compared with DGWO2, pGWO-CSA outperformed DGWO2 in seven out of fifteen test functions, and DGWO2 outperformed pGWO-CSA in test functions F1, F2, F3, F4, F6, F7, and F14.

In terms of the performance of the minimum. Sort by the number of optimal values. The pGWO-CSA and DGWO2 tied for first place with eight optimal values. OGWO ranked third with six optimal values. DGWO1 and GWO tied for fourth place with three optimal values. Compared with GWO, pGWO-CSA outperformed GWO in 12 out of 15 test functions. Compared with OGWO, pGWO-CSA outperformed OGWO in eight out of fifteen test functions, and OGWO outperformed pGWO-CSA only in test functions F4, F7, and F8. Compared with DGWO1, pGWO-CSA outperformed DGWO1 in 12 out of 15 test functions. Compared with DGWO2, pGWO-CSA outperformed DGWO2 in seven out of fifteen test functions, and DGWO2 outperformed pGWO-CSA only in test functions F1, F2, F3, F4, and F13.

In terms of the performance of the maximum. Sort by the number of optimal values. The pGWO-CSA and DGWO2 tied for first place with seven optimal values. OGWO ranked third with three optimal values. DGWO1 ranked fourth with two optimal values. GWO ranked fifth with one optimal value. Compared with GWO, pGWO-CSA outperformed GWO in 12 out of 15 test functions, and GWO outperformed pGWO-CSA only in test functions F7 and F15. Compared with OGWO, pGWO-CSA outperformed OGWO in 11 out of 15 test functions, and OGWO outperformed pGWO-CSA only in test functions F6 and F7. Compared with DGWO1, pGWO-CSA outperformed DGWO1 in 12 out of 15 test functions, and DGWO1 outperformed pGWO-CSA only in test functions F6 and F15. Compared with DGWO2, pGWO-CSA outperformed DGWO2 in eight out of fifteen test functions, and DGWO2 outperformed pGWO-CSA only in test functions F1, F2, F3, F4, F6, and F14.

In addition, pGWO-CSA can find theoretical optimal values on the test functions F9, F11, and F12. In the test functions F5, F9, F10, F11, and F12, pGWO-CSA is the optimal value among the five algorithms in terms of the mean, standard deviation, and minimum and maximum. Although DGWO2 outperformed pGWO-CSA in the performance of the first four test functions, F1, F2, F3, and F4, pGWO-CSA still outperformed the other three algorithms.

In terms of the performance of the mean. Sort by the number of optimal values. The pGWO-CSA ranked first with eight optimal values. DGWO2 ranked second with six optimal values. OGWO ranked third with three optimal values. DGWO1 and GWO tied for fourth place with one optimal value. Compared with GWO, pGWO-CSA outperformed GWO in 14 out of 15 test functions. Compared with OGWO, pGWO-CSA outperformed OGWO in 11 out of 15 test functions, and OGWO outperformed pGWO-CSA only in test functions F3, F7, and F10. Compared with DGWO1, pGWO-CSA outperformed DGWO1 in 14 out of 15 test functions. Compared with DGWO2, pGWO-CSA outperformed DGWO2 in nine out of fifteen test functions, and DGWO2 outperformed pGWO-CSA only in test functions F1, F2, F3, F4, and F14.

In terms of the performance of the standard deviation. Sort by the number of optimal values. The pGWO-CSA ranked first with eight optimal values. DGWO2 ranked second with seven optimal values. OGWO ranked third with two optimal values. GWO ranked fourth with one optimal value. DGWO1 ranked fifth with zero optimal values. Compared with GWO, pGWO-CSA outperformed GWO in 13 out of 15 test functions, and GWO only outperformed pGWO-CSA in test function F7. Compared with OGWO, pGWO-CSA outperformed OGWO in 11 out of 15 test functions, and OGWO only outperformed pGWO-CSA in test functions F3, F7, and F14. Compared with DGWO1, pGWO-CSA outperformed DGWO1 in all 15 test functions. Compared with DGWO2, pGWO-CSA outperformed DGWO2 in seven out of fifteen test functions, and DGWO2 outperformed pGWO-CSA in test functions F1, F2, F3, F4, F7, F10, and F14.

In terms of the performance of the minimum. Sort by the number of optimal values. DGWO2 ranked first with eight optimal values. The pGWO-CSA and OGWO tied for second place with six optimal values. GWO ranked fourth with two optimal values. DGWO1 ranked fifth with one optimal value. Compared with GWO, pGWO-CSA outperformed GWO in 13 out of 15 test functions. Compared with OGWO, pGWO-CSA outperformed OGWO in six out of fifteen test functions, and OGWO outperformed pGWO-CSA in test functions F1, F3, F4, F7, F10, and F13. Compared with DGWO1, pGWO-CSA outperformed DGWO1 in 14 out of 15 test functions, and DGWO1 outperformed pGWO-CSA only in test function F6. Compared with DGWO2, pGWO-CSA outperformed DGWO2 in six out of fifteen test functions, and DGWO2 outperformed pGWO-CSA in test functions F1, F2, F3, F4, F13, and F14.

In terms of the performance of the maximum. Sort by the number of optimal values. DGWO2 ranked first with eight optimal values. The pGWO-CSA ranked second with seven optimal values. OGWO ranked third with three optimal values. GWO ranked fourth with one optimal value. DGWO1 ranked fifth with zero optimal values. Compared with GWO, pGWO-CSA outperformed GWO in 12 out of 15 test functions, and GWO outperformed pGWO-CSA only in test functions F7 and F14. Compared with OGWO, pGWO-CSA outperformed OGWO in 10 out of 15 test functions, and OGWO outperformed pGWO-CSA only in test functions F4, F5, F7, and F14. Compared with DGWO1, pGWO-CSA outperformed DGWO1 in 13 out of 15 test functions, and DGWO1 outperformed pGWO-CSA only in test function F15. Compared with DGWO2, pGWO-CSA outperformed DGWO2 in six out of fifteen test functions, and DGWO2 outperformed pGWO-CSA in test functions F1, F2, F3, F4, F5, F7, F14, and F15.

In addition, pGWO-CSA can find theoretical optimal values on the test functions F9, F11, and F12. In the test functions F8, F9, F11, and F12, pGWO-CSA is the optimal value among the five algorithms in terms of the mean, standard deviation, and minimum and maximum. It is not difficult to find from Table 6 and Table 7 that DGWO2 performs better than pGWO-CSA in the first four test functions, F1–F4, in both the 30 dimensions and the 50 dimensions. As can be seen from Table 1, the first four test functions are simple single-peak functions, indicating that DGWO1 performs better than pGWO-CSA in simple single-peak functions. However, compared with the other three algorithms, pGWO-CSA still performs better on the test functions F1–F4.

By comparison, it is not difficult to find that pGWO-CSA performs better than the previous seven test functions, F1–F7, in the following eight test functions, F8–F15, whether in the 30 dimensions or the 50 dimensions. It can be seen that pGWO-CSA performs better in more complex functions, which is largely due to the super-mutation operation carried out by pGWO-CSA, which helps pGWO-CSA better jump out of local optimum.

Based on the above data and analysis, pGWO-CSA has faster convergence speed, higher accuracy, and better ability to jump out of the local optimum compared with GWO and its variants in either 30 or 50 dimensions. In order to further reveal the performance of pGWO-CSA, the Wilcoxon test is performed in Section 4.3 based on the experimental data in Section 4.1 and Section 4.2.

### 4.3. Wilcoxon Test

In order to further reveal the performance of pGWO-CSA, according to the experimental data in Section 4.1 and Section 4.2, the Wilcoxon test is conducted on the mean of the 30 running results of each algorithm. The statistical results are shown in Table 8. In the Wilcoxon test, ’+’ means that the proposed algorithm is inferior to the selected algorithm, ‘−‘ means that the proposed algorithm is superior to the selected algorithm, and ‘=’ means that the two algorithms get the same result.

It can be seen from Table 8 that the number of ‘+’ of each algorithm is small, indicating that the seven algorithms compared with pGWO-CSA only outperform pGWO-CSA in a few test functions, and the number of ‘−’ of each algorithm exceeds 15, indicating that pGWO-CSA outperformed other algorithms in most test functions. The results show that pGWO-CSA is superior to other swarm intelligence algorithms, GWO, and its variants.

## 5. Robot Path-Planning Problem

With the development of artificial intelligence, robots have been widely used in various fields [33,34,35]. Among them, robot path planning is an important research problem. To further verify the applicability and superiority of the proposed algorithm, it is applied to the robot path-planning problem.

### 5.1. Robot Path-Planning Problem Description

The robot path-planning problem mainly includes two aspects: environment modeling and evaluation function. Environment modeling is to transform the environmental information of the robot into a form that can be recognized and expressed by a computer. The evaluation function is used to measure the path quality and is regarded as the objective function to be optimized by the algorithm.

#### 5.1.1. Environment Modeling

The environment model of the robot path-planning problem is shown in Figure 35. The starting point is located at (0,0) and marked with a black star, the endpoint is located at (10,10) and marked with a blue triangle, and the obstacles are marked with a green circle. The mathematical expression of the obstacles is shown in Formula (13).
(13)$${\left(x-a\right)}^{2}+{\left(y-b\right)}^{2}=r^{2},$$
where *a* and *b* represent the center coordinates of the obstacle, and *r* represents the radius of the circle.

#### 5.1.2. Evaluation Function

Suppose the robot finds some path points from start to end: ($x_{0}$, $y_{0}$), ($x_{1}$, $y_{1}$), …, ($x_{n}$, $y_{n}$), and the coordinate of the path $point_{i}$ is ($x_{i}$, $y_{i}$). A complete path formed by connecting these path points is a feasible solution to the robot path-planning problem. In order to reduce the optimization dimension of the problem and smooth the path curve, the spline interpolation method is used to construct the path curve. In order to evaluate the quality of the path, this paper considers the length of the path and the risk of the path. The evaluation function is shown in Formula (14).
(14)$$fit=w1\times fit_{len}+w2\times fit_{risk},$$
where *w1* and *w2* are weight parameters and *w*1 + *w*2 = 1.0. $fit_{len}$ represents the fitness value of the length of the path, which is calculated by Formula (15); $fit_{rick}$ represents the fitness value of the risk of the path, which is calculated by Formula (16).
(15)$$fit_{len}=\overset{n}{\underset{i=1}{\sum }}\sqrt{{\left(x_{i}-x_{i-1}\right)}^{2}+{\left(y_{i}-y_{i-1}\right)}^{2}},$$
where *n* is the total number of path points, and ($x_{i}$, $y_{i}$) represents the coordinate of the path $point_{i}$.
(16)$$fit_{risk}=c\times \sum_{i=1}^{k}\sum_{j=1}^{n}\max\left(0,\;1-\frac{\sqrt{{\left(a_{i}-x_{j}\right)}^{2}+{\left(b_{i}-y_{j}\right)}^{2}}}{r_{i}}\right),$$
where *c* is the penalty coefficient, *k* is the total number of obstacles, ($a_{i}$, $b_{i}$) is the coordinates of the center of obstacle *i*, and $r_{i}$ is the radius.

According to Formulas (14)–(16), when the fitness value of $fit_{len}$ is small, then the length of the path is short. When the fitness value of $fit_{risk}$ is small, the risk of the path is low. Therefore, the smaller the fit, the higher the quality of the path.

### 5.2. The Experimental Results

In order to verify the applicability and superiority of pGWO-CSA in robot path-planning problems, PSO, DE, FA, GWO, and its variants are compared with pGWO-CSA. The parameters of all algorithms are exactly the same as in Section 4. In order to avoid the randomness of the algorithm, each algorithm will be run 10 times, and then the minimum, maximum, and mean of the results will be recorded. The path planned by pGWO-CSA is shown in Figure 36, and the experimental results of all algorithms are shown in Table 9.

According to the experimental data, pGWO-CSA is the optimal value of all algorithms in the performance of the minimum, maximum, and mean. The applicability and superiority of pGWO-CSA are further verified.

## 6. Conclusions

Aiming at the defects of the GWO, such as low convergence accuracy and easy precocity when dealing with complex problems, this paper proposes pGWO-CSA to settle these drawbacks. Firstly, the pGWO-CSA uses a nonlinear function instead of a linear function to adjust the iterative attenuation of the convergence factor to balance exploitation and exploration. Secondly, pGWO-CSA improves GWO’s position-updating strategy, and finally, pGWO-CSA is mixed with the CSA. The improved pGWO-CSA improves the convergence speed, precision, and ability to jump out of the local optimum. The experimental results show that the pGWO-CSA has obvious accuracy advantages. Compared with GWO and its variants participating in the experiment, the pGWO-CSA shows good stability in both 30 and 50 dimensions and is suitable for the optimization of complex and variable problems. Finally, the proposed algorithm is applied to the robot path-planning problem, which further verifies the applicability and superiority of the proposed algorithm.

## Figures and Tables

**Figure 1 biomimetics-08-00084-f001:**
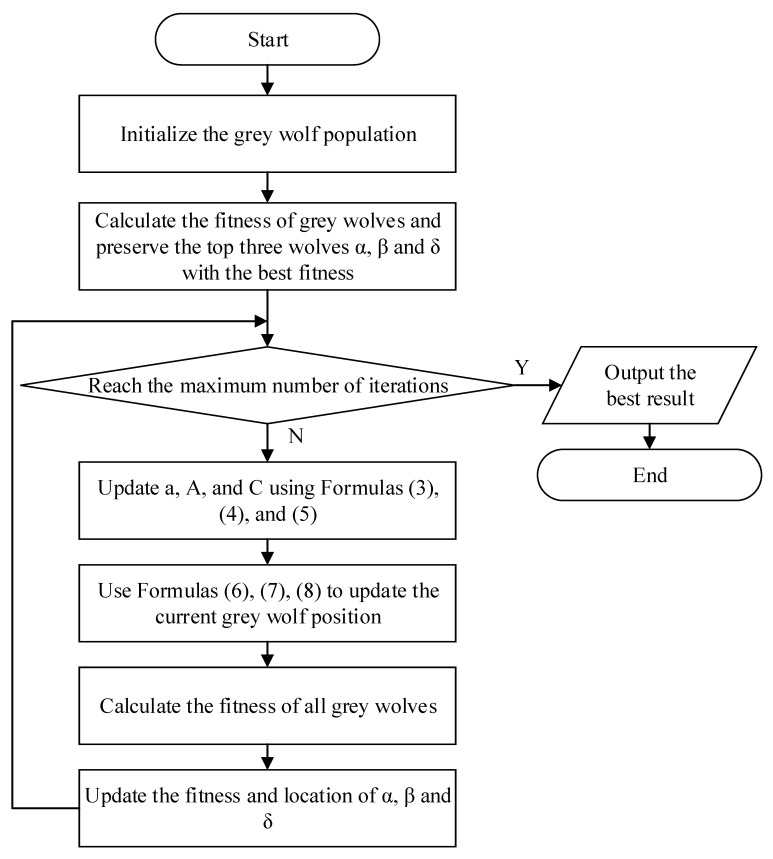
Flow chart of GWO algorithm.

**Figure 2 biomimetics-08-00084-f002:**
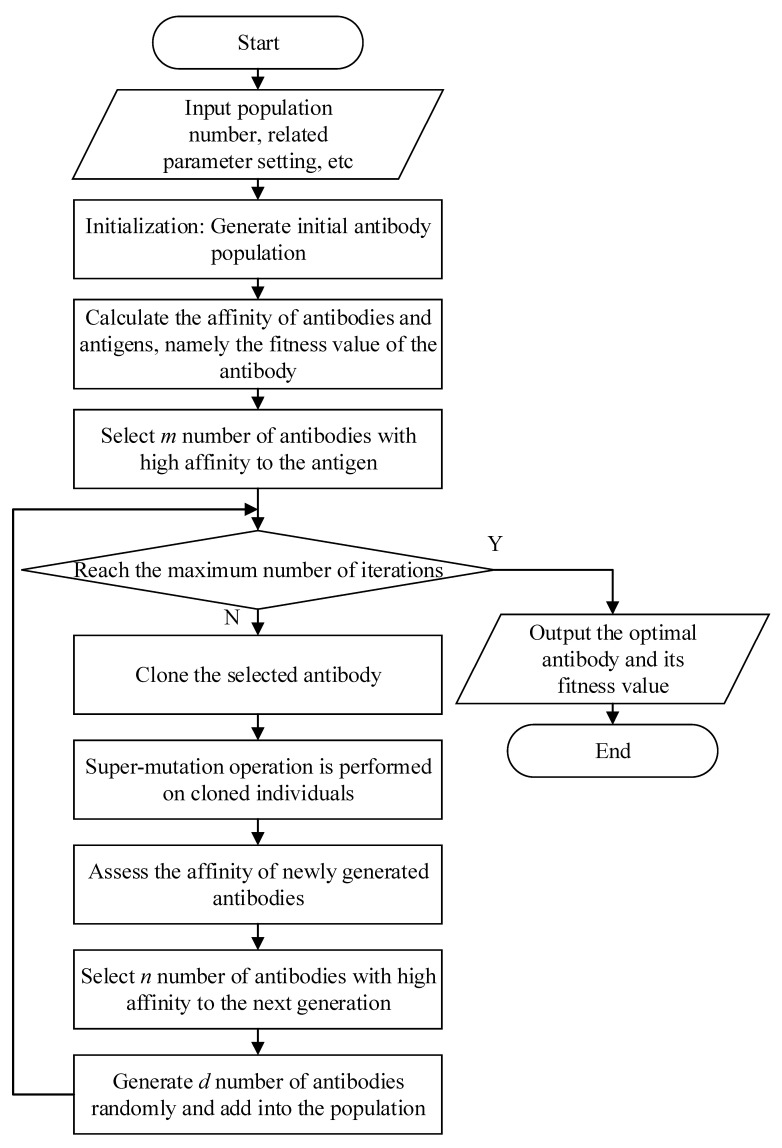
Flow chart of CSA.

**Figure 3 biomimetics-08-00084-f003:**
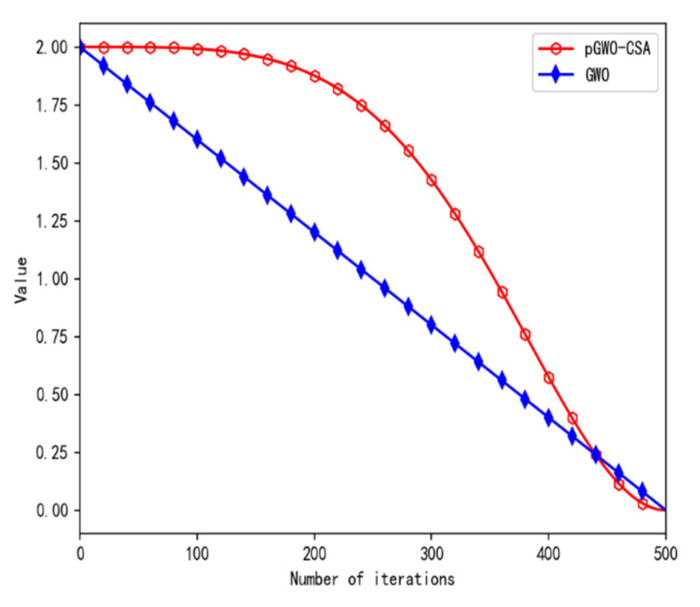
Iterative curve of parameter a.

**Figure 4 biomimetics-08-00084-f004:**
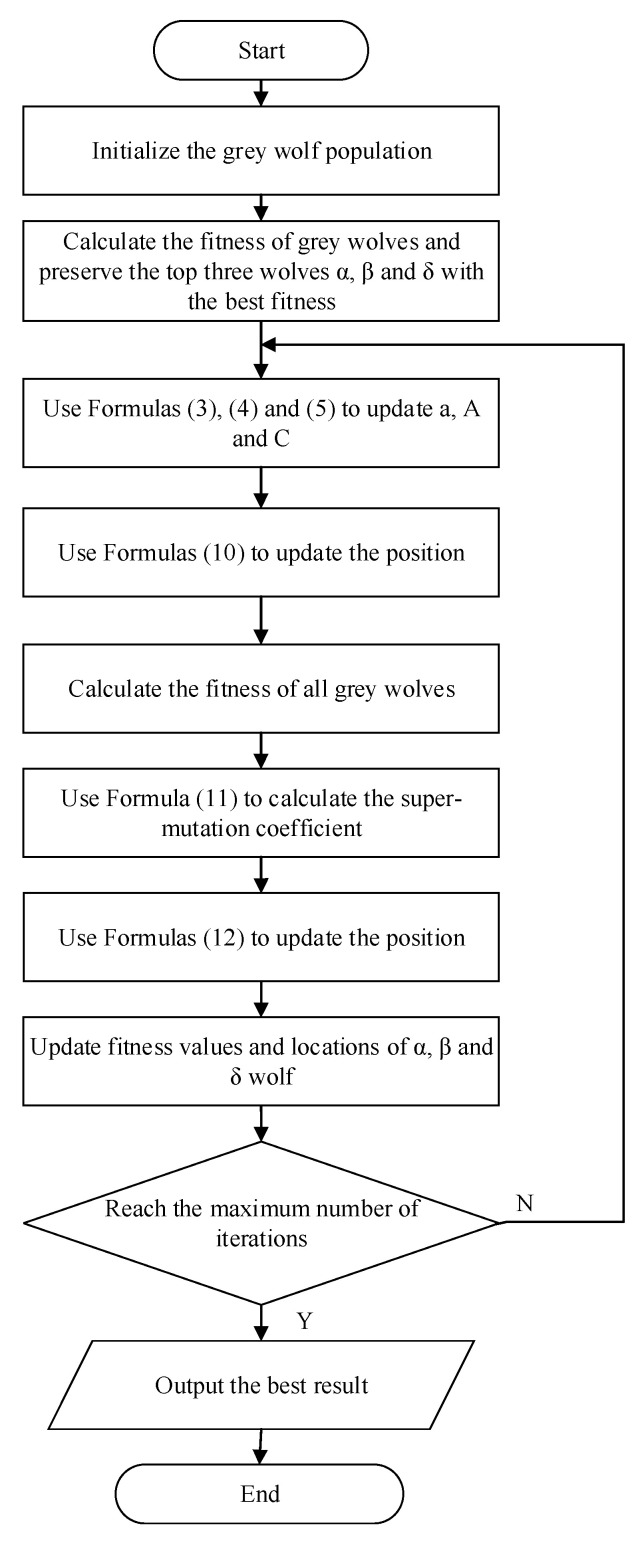
Flow chart of pGWO-CSA.

**Figure 5 biomimetics-08-00084-f005:**
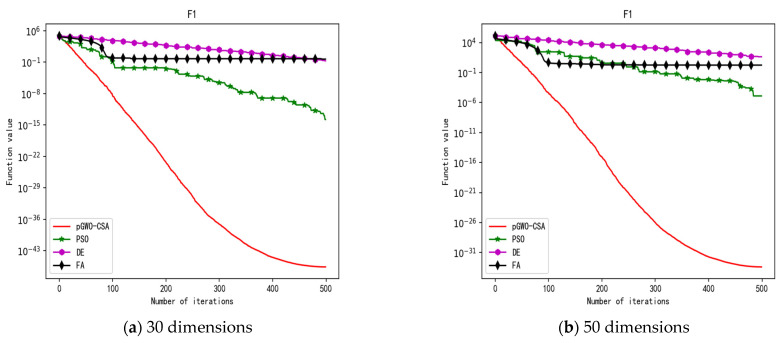
Convergence curve of test function F1.

**Figure 6 biomimetics-08-00084-f006:**
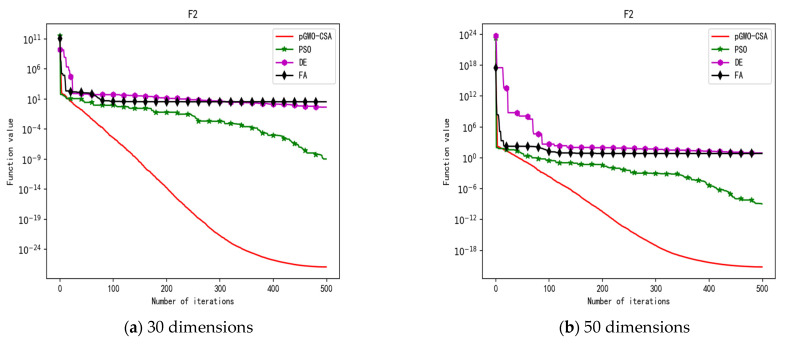
Convergence curve of test function F2.

**Figure 7 biomimetics-08-00084-f007:**
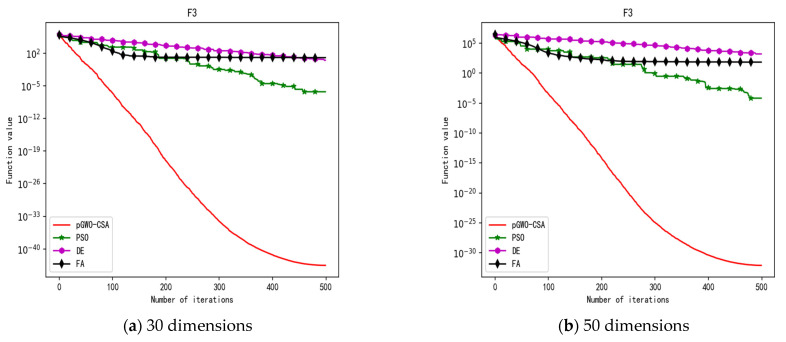
Convergence curve of test function F3.

**Figure 8 biomimetics-08-00084-f008:**
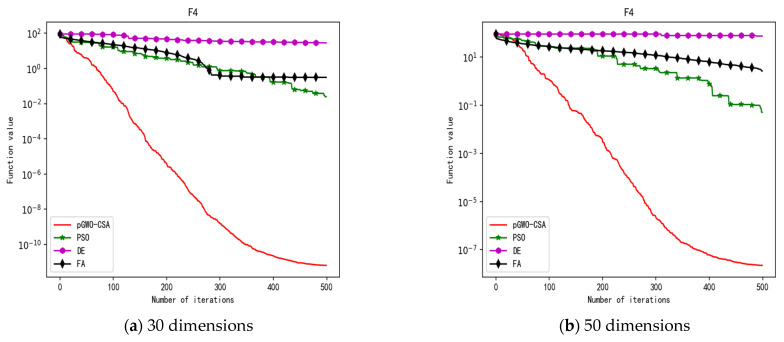
Convergence curve of test function F4.

**Figure 9 biomimetics-08-00084-f009:**
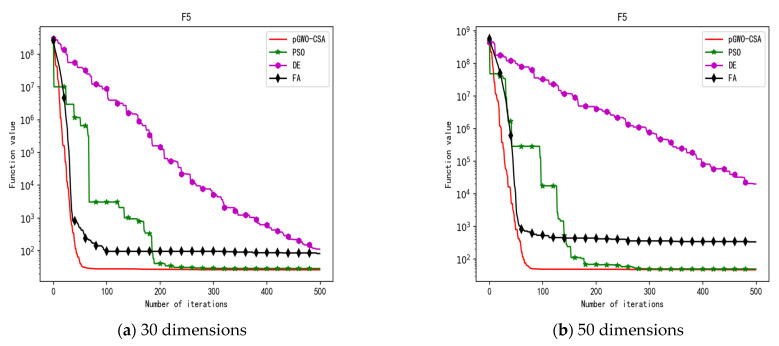
Convergence curve of test function F5.

**Figure 10 biomimetics-08-00084-f010:**
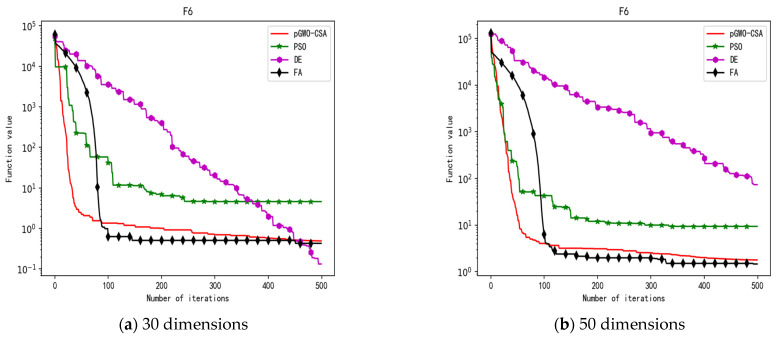
Convergence curve of test function F6.

**Figure 11 biomimetics-08-00084-f011:**
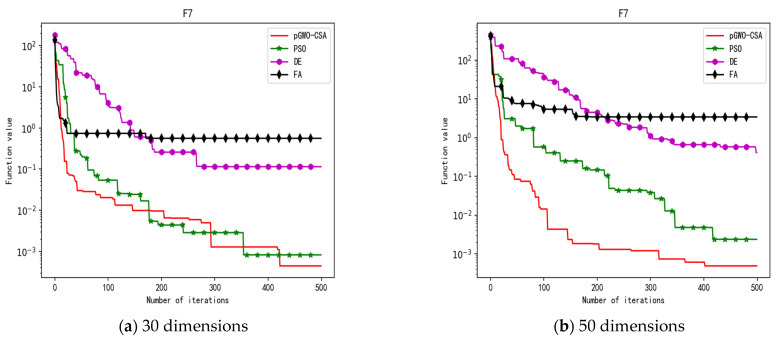
Convergence curve of test function F7.

**Figure 12 biomimetics-08-00084-f012:**
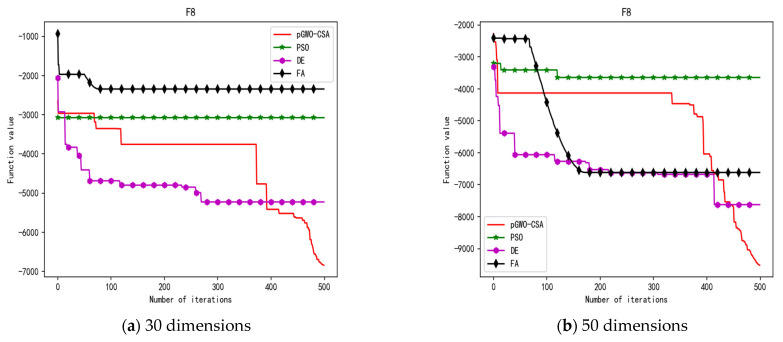
Convergence curve of test function F8.

**Figure 13 biomimetics-08-00084-f013:**
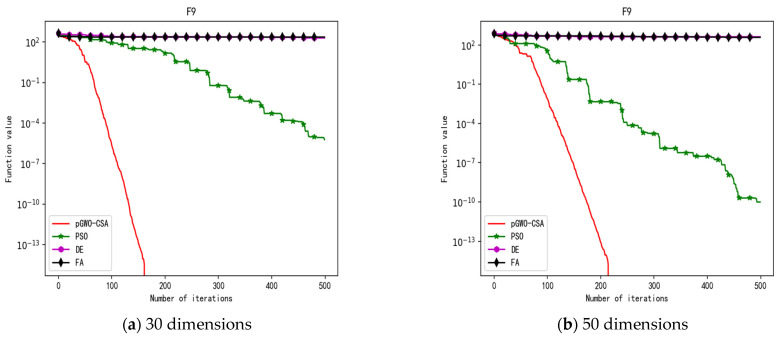
Convergence curve of test function F9.

**Figure 14 biomimetics-08-00084-f014:**
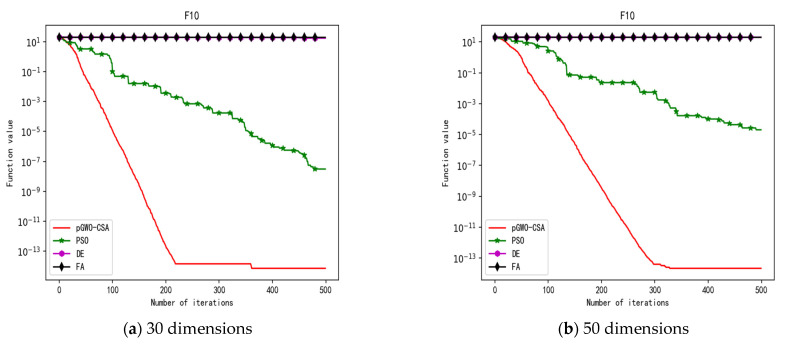
Convergence curve of test function F10.

**Figure 15 biomimetics-08-00084-f015:**
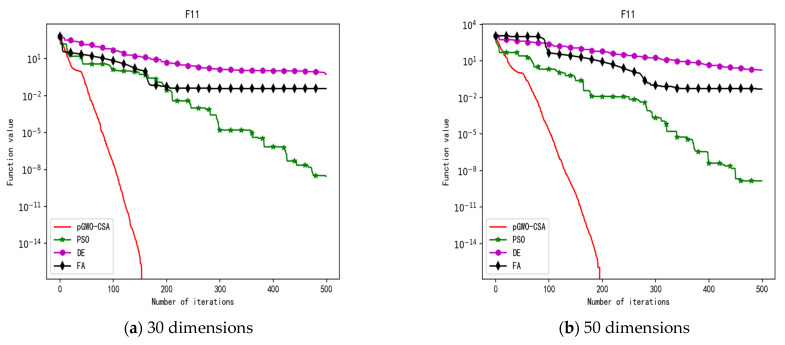
Convergence curve of test function F11.

**Figure 16 biomimetics-08-00084-f016:**
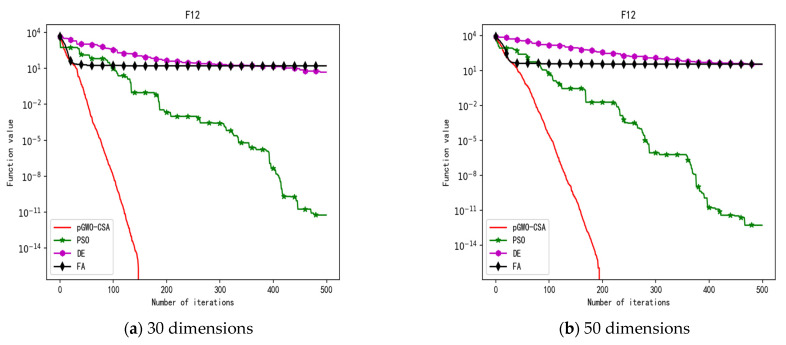
Convergence curve of test function F12.

**Figure 17 biomimetics-08-00084-f017:**
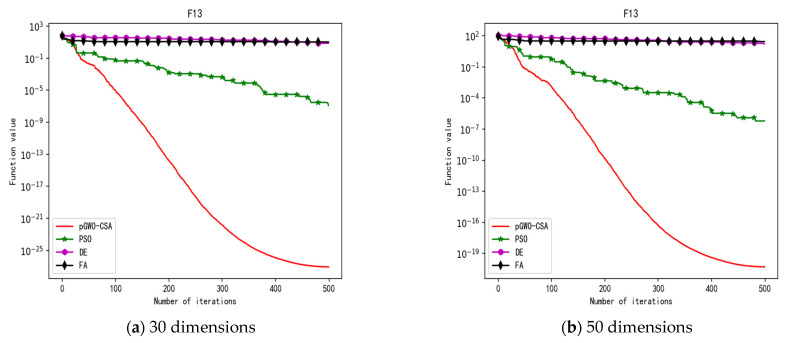
Convergence curve of test function F13.

**Figure 18 biomimetics-08-00084-f018:**
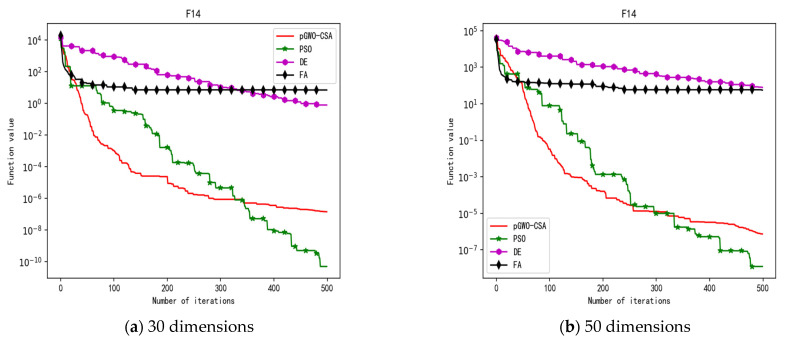
Convergence curve of test function F14.

**Figure 19 biomimetics-08-00084-f019:**
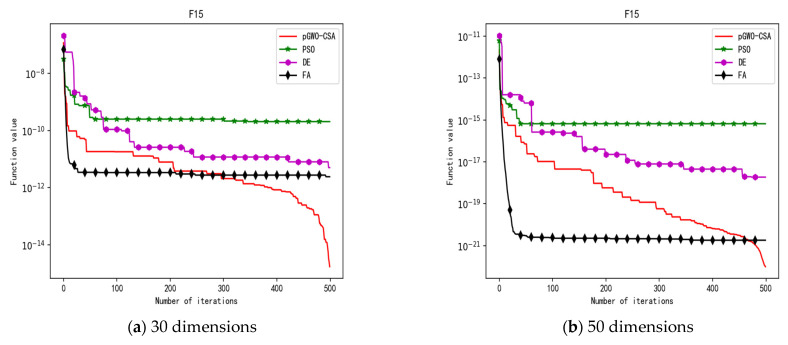
Convergence curve of test function F15.

**Figure 20 biomimetics-08-00084-f020:**
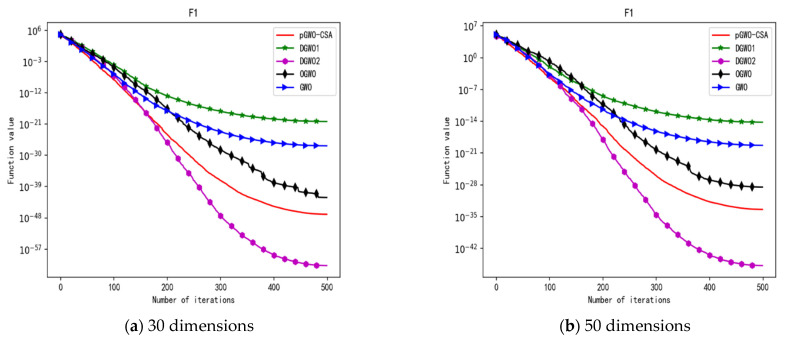
Convergence curve of test function F1.

**Figure 21 biomimetics-08-00084-f021:**
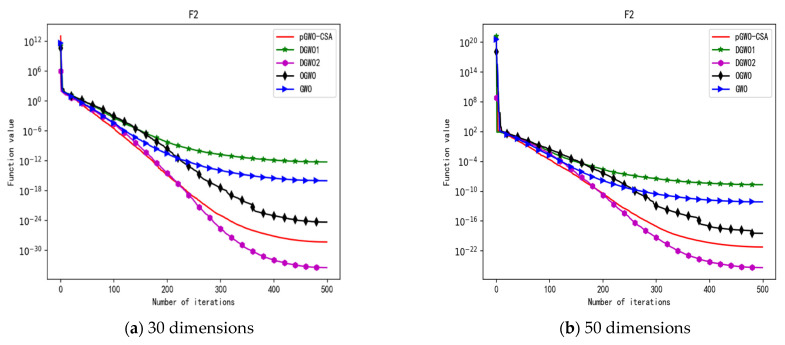
Convergence curve of test function F2.

**Figure 22 biomimetics-08-00084-f022:**
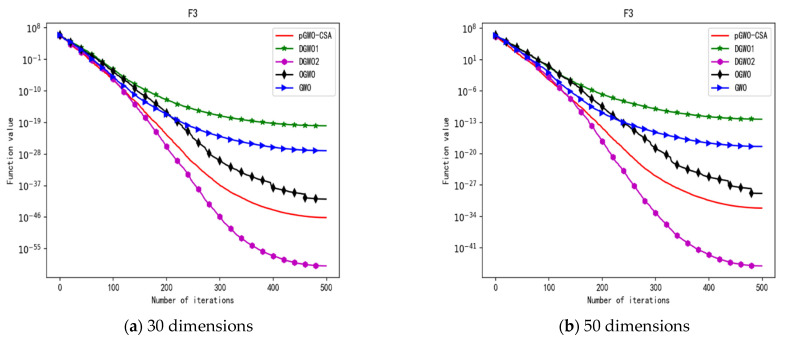
Convergence curve of test function F3.

**Figure 23 biomimetics-08-00084-f023:**
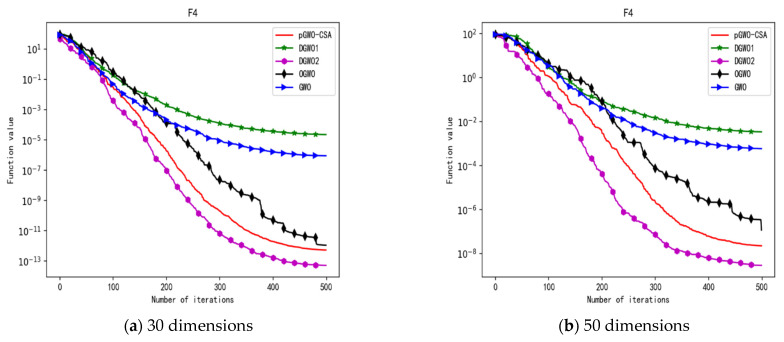
Convergence curve of test function F4.

**Figure 24 biomimetics-08-00084-f024:**
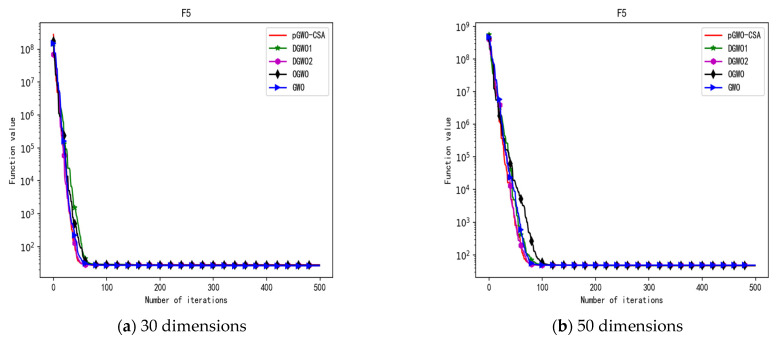
Convergence curve of test function F5.

**Figure 25 biomimetics-08-00084-f025:**
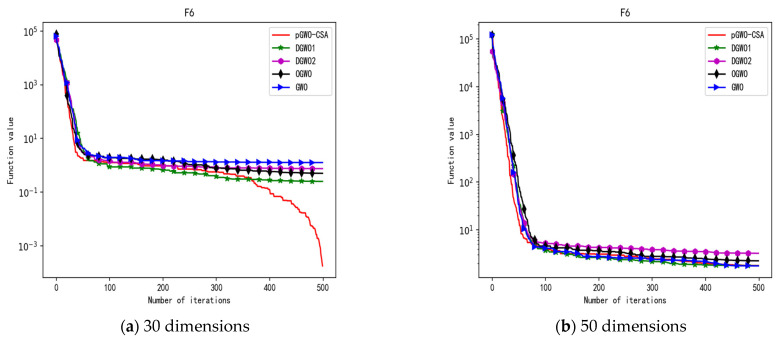
Convergence curve of test function F6.

**Figure 26 biomimetics-08-00084-f026:**
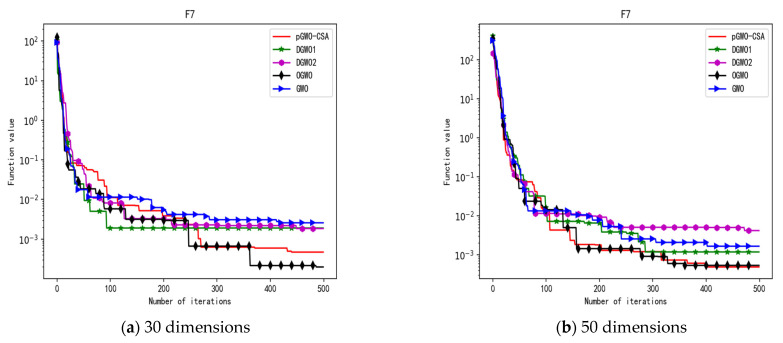
Convergence curve of test function F7.

**Figure 27 biomimetics-08-00084-f027:**
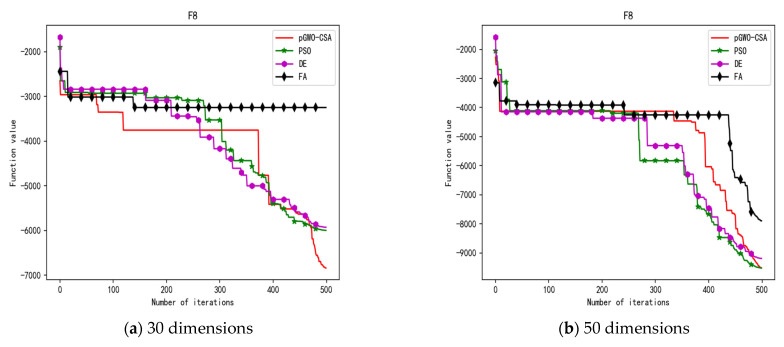
Convergence curve of test function F8.

**Figure 28 biomimetics-08-00084-f028:**
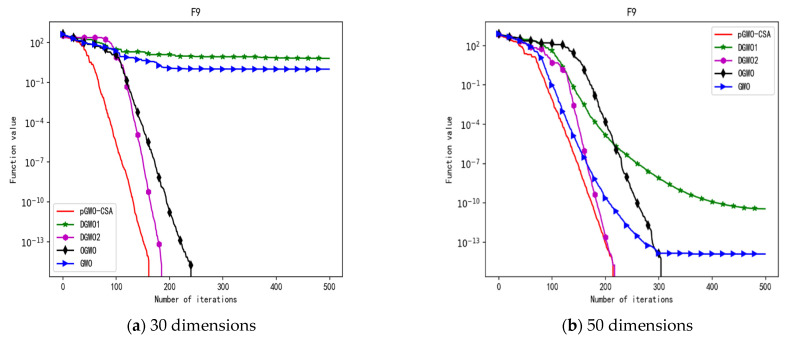
Convergence curve of test function F9.

**Figure 29 biomimetics-08-00084-f029:**
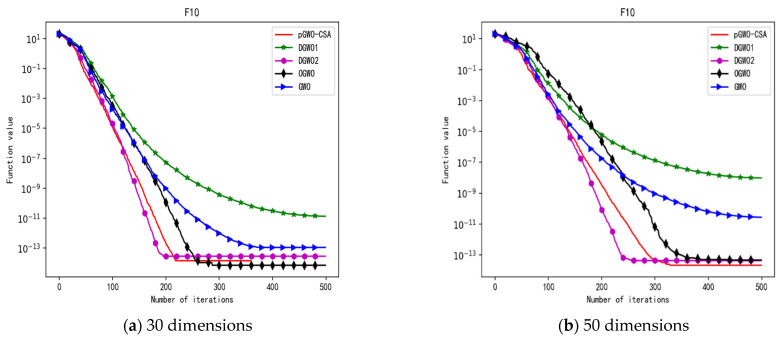
Convergence curve of test function F10.

**Figure 30 biomimetics-08-00084-f030:**
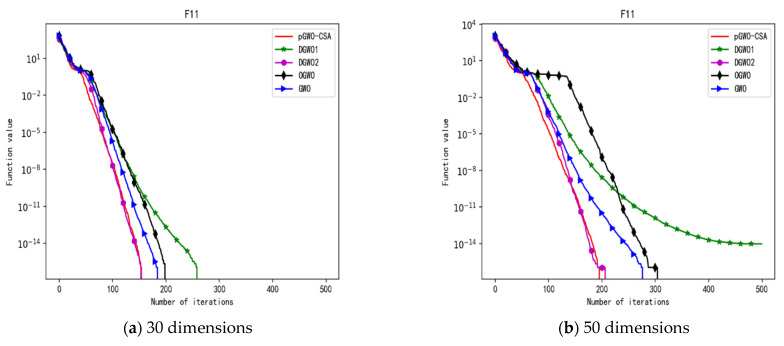
Convergence curve of test function F11.

**Figure 31 biomimetics-08-00084-f031:**
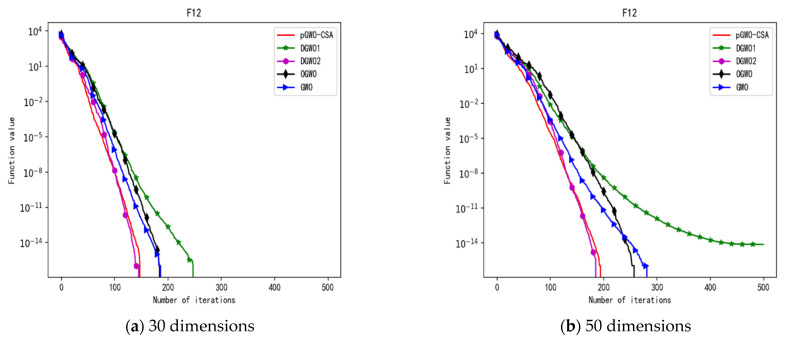
Convergence curve of test function F12.

**Figure 32 biomimetics-08-00084-f032:**
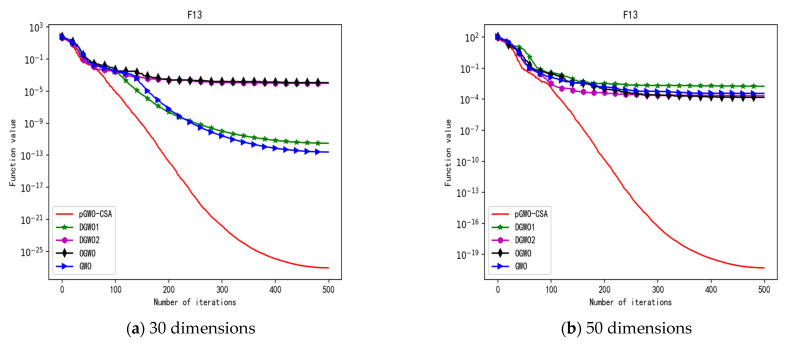
Convergence curve of test function F12.

**Figure 33 biomimetics-08-00084-f033:**
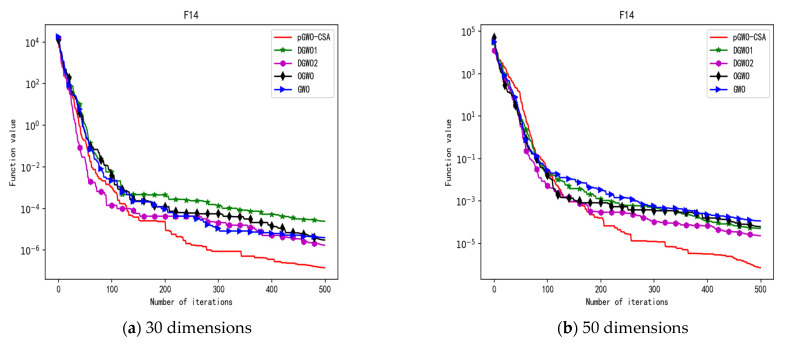
Convergence curve of test function F14.

**Figure 34 biomimetics-08-00084-f034:**
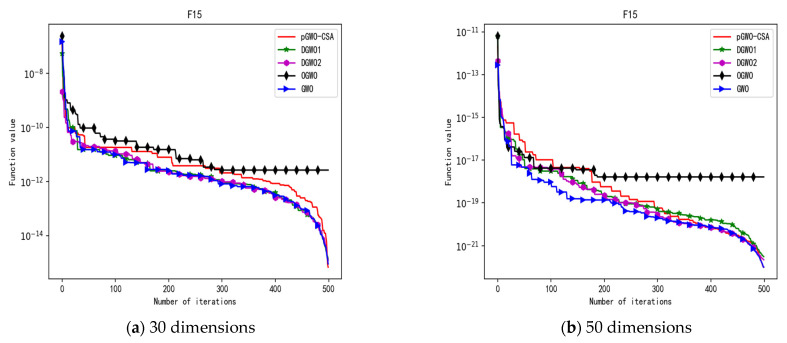
Convergence curve of test function F15.

**Figure 35 biomimetics-08-00084-f035:**
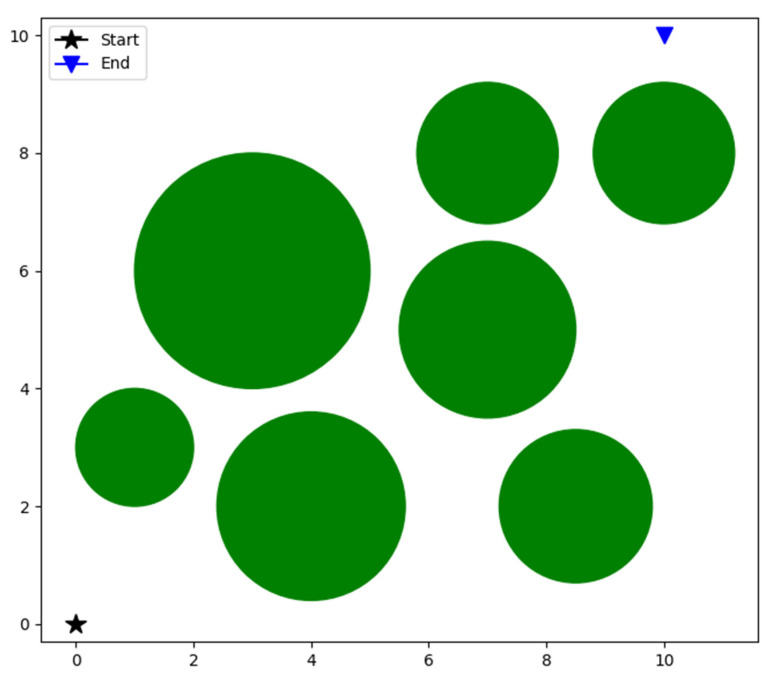
Environment modeling.

**Figure 36 biomimetics-08-00084-f036:**
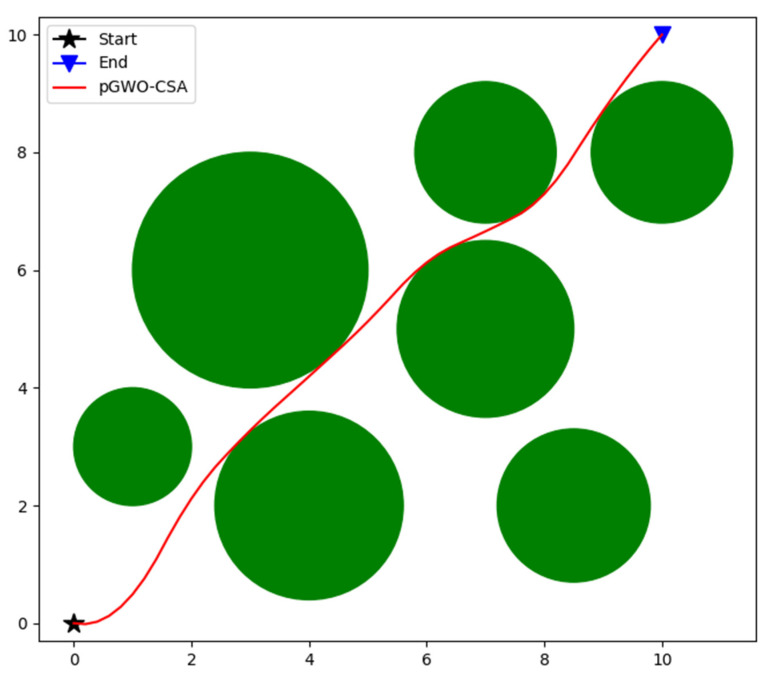
The path of pGWO-CSA.

**Table 1 biomimetics-08-00084-t001:** Detailed description of the test functions F1–F15.

No.	Function	Dimension	Interval	*f_min_*
F1	$f\left(x\right)=\sum_{i=1}^{d}x_{i}{}^{2}$	30, 50	[−100, 100]	0
F2	$f\left(x\right)=\sum_{i=1}^{d}\left|x_{i}\right|+\prod_{i=1}^{d}\left|x_{i}\right|$	30, 50	[−10, 10]	0
F3	$f\left(x\right)=\sum_{i=1}^{d}\left(\sum_{j=1}^{i}x_{j}^{2}\right)$	30, 50	[−100, 100]	0
F4	$f\left(x\right)=\max_{i}\left\{\left|x_{i\;}\right|,1\leq i\leq n\right\}$	30, 50	[−100, 100]	0
F5	$f\left(x\right)=\sum_{i=1}^{d-1}\left[100{\left(x_{i+1}-x_{i}{}^{2}\right)}^{2}+{\left(x_{i}-1\right)}^{2}\right]$	30, 50	[−30, 30]	0
F6	$f\left(x\right)=\sum_{i=1}^{d}{\left(\left[x_{i}+0.5\right]\right)}^{2}$	30, 50	[−100, 100]	0
F7	$f\left(x\right)=\sum_{i=1}^{d}ix_{i}{}^{4}+random\left(0,1\right)$	30, 50	[−1.28, 1.28]	0
F8	$f\left(x\right)=\sum_{i=1}^{d}-x_{i}\sin\left(\sqrt{\left|x_{i}\right|}\right)$	30, 50	[−500, 500]	−418.9829 × 5
F9	$f\left(x\right)=\sum_{i=1}^{d}\left[x_{i}{}^{2}-10\cos\left(2\Pi x_{i}\right)+10\right]$	30, 50	[−5.12, 5.12]	0
F10	$\begin{array}{l} f\left(x\right)=-20\exp\left(-0.2\sqrt{\frac{1}{n}\sum_{i=1}^{d}x_{i}{}^{2}}\right)-\\ \;\exp\left(\frac{1}{n}\sum_{i=1}^{d}\cos\left(2\Pi x_{i}\right)\right)+20+e \end{array}$	30, 50	[−32, 32]	0
F11	$f\left(x\right)=\frac{1}{4000}\sum_{i=1}^{d}x_{i}{}^{2}-\prod_{i=1}^{d}\cos\left(\frac{x_{i}}{\sqrt{i}}\right)+1$	30, 50	[−600, 600]	0
F12	$f(x)=\sum_{i=1}^{d-1}\left[\begin{array}{c} x_{i}^{2}+2x_{i+1}^{2}-0.3\cos\left(3\pi x_{i}\right)\\ -0.4\cos\left(4\pi x_{i+1}\right)+0.7 \end{array}\right]$	30, 50	[−15, 15]	0
F13	$f(x)=\sum_{i=1}^{d}|x_{i}\sin\left(x_{i}\right)+0.1x_{i}|$	30, 50	[−10, 10]	0
F14	$f(x)=\sum_{i=1}^{d/4}\left[\begin{array}{c} {\left(x_{4i-3}+10x_{4i-2}\right)}^{2}+5{\left(x_{4i-1}-x_{4i}\right)}^{2}\\ +{\left(x_{4i-2}-2x_{4i-1}\right)}^{4}+10{\left(x_{4i-3}-x_{4i}\right)}^{4} \end{array}\right]$	30, 50	[−4, 5]	0
F15	$\begin{array}{l} f(x)=\left\{\left[\sum_{i=1}^{d}\sin^{2}\left(x_{i}\right)\right]-exp\left(-\sum_{i=1}^{d}x_{i}{}^{2}\right)\right\}\\ \cdot exp\left[-\sum_{i=1}^{d}\sin^{2}\sqrt{\left|x_{i}\right|}\right] \end{array}$	30, 50	[−10, 10]	−1

**Table 2 biomimetics-08-00084-t002:** Main parameters of the four algorithms.

Algorithm	The Main Parameters
PSO	ω = 0.8, c1 = 1.5, c2 = 1.5
DE	CR = 0.8, F = 0.6
FA	*β*0 = 1.0, γ = 0.000001, *α* = 0.6
pGWO-CSA	$\vec{a}$ nonlinearly decreases from 2 to 0, u = 2

**Table 3 biomimetics-08-00084-t003:** The experimental results under 30 dimensions.

Function	Index	PSO	DE	FA	pGWO-CSA
F1	mean	6.18 × 10^−10^	2.35 × 10^−1^	4.92 × 10^−1^	5.97 × 10^−44^
	std	1.97 × 10^−9^	1.41 × 10^−1^	6.87 × 10^−2^	1.12 × 10^−43^
	min	5.49 × 10^−18^	4.00 × 10^−2^	3.08 × 10^−1^	6.65 × 10^−47^
	max	1.09 × 10^−8^	6.23 × 10^−1^	6.09 × 10^−1^	4.87 × 10^−43^
F2	mean	4.30 × 10^−7^	3.48 × 10^−1^	3.19 × 10^0^	3.66 × 10^−27^
	std	1.70 × 10^−6^	1.13 × 10^−1^	1.65 × 10^−1^	3.51 × 10^−27^
	min	8.23 × 10^−13^	1.87 × 10^−1^	2.93 × 10^0^	3.98 × 10^−28^
	max	9.52 × 10^−6^	7.07 × 10^−1^	3.53 × 10^0^	1.54 × 10^−26^
F3	mean	1.12 × 10^−6^	3.45 × 10^0^	8.68 × 10^0^	2.24 × 10^−42^
	std	5.03 × 10^−6^	1.97 × 10^0^	1.32 × 10^0^	8.92 × 10^−42^
	min	2.17 × 10^−16^	6.70 × 10^−1^	6.05 × 10^0^	3.59 × 10^−45^
	max	2.78 × 10^−5^	8.94 × 10^0^	1.18 × 10^1^	5.00 × 10^−41^
F4	mean	3.24 × 10^−3^	2.09 × 10^1^	3.12 × 10^−1^	1.08 × 10^−11^
	std	5.28 × 10^−3^	6.64 × 10^0^	2.14 × 10^−2^	1.21 × 10^−11^
	min	8.06 × 10^−7^	1.13 × 10^1^	2.72 × 10^−1^	1.00 × 10^−12^
	max	2.34 × 10^−2^	4.23 × 10^1^	3.49 × 10^−1^	4.94 × 10^−11^
F5	mean	2.82 × 10^1^	1.73 × 10^2^	1.63 × 10^2^	2.67 × 10^1^
	std	4.07 × 10^−1^	9.13 × 10^1^	1.64 × 10^2^	5.18 × 10^−1^
	min	2.75 × 10^1^	4.80 × 10^1^	6.81 × 10^1^	2.59 × 10^1^
	max	2.89 × 10^1^	4.90 × 10^2^	6.18 × 10^2^	2.80 × 10^1^
F6	mean	4.81 × 10^0^	2.11 × 10^−1^	5.06 × 10^−1^	4.20 × 10^−1^
	std	1.61 × 10^−1^	1.21 × 10^−1^	3.58 × 10^−2^	3.28 × 10^−1^
	min	4.19 × 10^0^	3.27 × 10^−2^	3.93 × 10^−1^	3.14 × 10^−6^
	max	5.05 × 10^0^	4.90 × 10^−1^	5.62 × 10^−1^	1.50 × 10^0^
F7	mean	2.36 × 10^−3^	1.08 × 10^−1^	5.54 × 10^−1^	1.38 × 10^−3^
	std	2.06 × 10^−3^	2.78 × 10^−2^	1.08 × 10^−1^	8.15 × 10^−4^
	min	1.19 × 10^−4^	4.20 × 10^−2^	2.92 × 10^−1^	3.62 × 10^−4^
	max	1.01 × 10^−2^	1.71 × 10^−1^	8.16 × 10^−1^	3.67 × 10^−3^
F8	mean	−2.98 × 10^3^	−5.88 × 10^3^	−4.16 × 10^3^	−6.13 × 10^3^
	std	3.37 × 10^2^	4.45 × 10^2^	1.45 × 10^3^	7.70 × 10^2^
	min	−3.79 × 10^3^	−7.27 × 10^3^	−7.10 × 10^3^	−7.55 × 10^3^
	max	−2.52 × 10^3^	−5.15 × 10^3^	−2.14 × 10^3^	−4.68 × 10^3^
F9	mean	2.18 × 10^−7^	2.07 × 10^2^	2.28 × 10^2^	0.00 × 10^0^
	std	1.05 × 10^−6^	1.79 × 10^1^	2.52 × 10^1^	0.00 × 10^0^
	min	0.00 × 10^0^	1.59 × 10^2^	1.67 × 10^2^	0.00 × 10^0^
	max	5.88 × 10^−6^	2.57 × 10^2^	2.89 × 10^2^	0.00 × 10^0^
F10	mean	8.55 × 10^−6^	7.65 × 10^0^	1.96 × 10^1^	8.17 × 10^−15^
	std	2.19 × 10^−5^	9.03 × 10^0^	2.19 × 10^−1^	2.62 × 10^−15^
	min	2.69 × 10^−11^	1.16 × 10^−1^	1.89 × 10^1^	3.55 × 10^−15^
	max	1.19 × 10^−4^	2.00 × 10^1^	2.00 × 10^1^	1.42 × 10^−14^
F11	mean	1.36 × 10^−7^	5.09 × 10^−1^	4.10 × 10^−2^	0.00 × 10^0^
	std	4.09 × 10^−7^	2.14 × 10^−1^	1.68 × 10^−2^	0.00 × 10^0^
	min	3.33 × 10^−15^	1.09 × 10^−1^	2.01 × 10^−2^	0.00 × 10^0^
	max	1.73 × 10^−6^	9.29 × 10^−1^	9.22 × 10^−2^	0.00 × 10^0^
F12	mean	5.12 × 10^−8^	1.88 × 10^0^	1.57 × 10^1^	0.00 × 10^0^
	std	2.09 × 10^−7^	1.54 × 10^0^	1.26 × 10^0^	0.00 × 10^0^
	min	0.00 × 10^0^	2.85 × 10^−1^	1.21 × 10^1^	0.00 × 10^0^
	max	1.16 × 10^−6^	7.37 × 10^0^	1.74 × 10^1^	0.00 × 10^0^
F13	mean	2.04 × 10^−6^	4.98 × 10^0^	1.25 × 10^1^	1.28 × 10^−23^
	std	9.07 × 10^−6^	4.04 × 10^0^	2.34 × 10^0^	3.86 × 10^−23^
	min	1.42 × 10^−11^	6.93 × 10^−2^	7.89 × 10^0^	1.70 × 10^−27^
	max	5.07 × 10^−5^	1.36 × 10^1^	1.89 × 10^1^	1.68 × 10^−22^
F14	mean	2.15 × 10^−7^	1.28 × 10^0^	8.11 × 10^0^	5.36 × 10^−6^
	std	7.67 × 10^−7^	9.24 × 10^−1^	2.43 × 10^0^	7.72 × 10^−6^
	min	1.82 × 10^−17^	3.42 × 10^−1^	4.29 × 10^0^	2.86 × 10^−8^
	max	4.12 × 10^−6^	4.46 × 10^0^	1.38 × 10^1^	2.89 × 10^−5^
F15	mean	3.18 × 10^−10^	1.78 × 10^−11^	7.03 × 10^−12^	1.75 × 10^−16^
	std	1.38 × 10^−10^	1.04 × 10^−11^	1.01 × 10^−11^	5.99 × 10^−16^
	min	5.84 × 10^−11^	2.91 × 10^−12^	1.29 × 10^−13^	2.30 × 10^−17^
	max	6.33 × 10^−10^	4.72 × 10^−11^	4.54 × 10^−11^	3.39 × 10^−15^

**Table 4 biomimetics-08-00084-t004:** The experimental results under 50 dimensions.

Function	Index	PSO	DE	FA	pGWO–CSA
F1	mean	1.31 × 10^−7^	6.52 × 10^1^	1.51 × 10^0^	3.29 × 10^−31^
	std	2.80 × 10^−7^	2.88 × 10^1^	1.84 × 10^−1^	3.21 × 10^−31^
	min	7.01 × 10^−15^	2.19 × 10^1^	1.17 × 10^0^	2.68 × 10^−33^
	max	1.30 × 10^−6^	1.27 × 10^2^	1.83 × 10^0^	1.33 × 10^−30^
F2	mean	5.42 × 10^−6^	8.15 × 10^0^	2.03 × 10^1^	7.23 × 10^−20^
	std	2.41 × 10^−5^	4.81 × 10^0^	4.57 × 10^1^	4.78 × 10^−20^
	min	1.62 × 10^−10^	3.68 × 10^0^	6.38 × 10^0^	9.01 × 10^−21^
	max	1.35 × 10^−4^	3.09 × 10^1^	1.97 × 10^2^	1.95 × 10^−19^
F3	mean	2.28 × 10^−4^	1.28 × 10^3^	6.61 × 10^1^	6.91 × 10^−30^
	std	1.05 × 10^−3^	6.20 × 10^2^	1.27 × 10^1^	7.60 × 10^−30^
	min	2.86 × 10^−14^	5.03 × 10^2^	3.62 × 10^1^	1.26 × 10^−31^
	max	5.83 × 10^−3^	3.34 × 10^3^	8.97 × 10^1^	2.58 × 10^−29^
F4	mean	1.10 × 10^−1^	8.85 × 10^1^	3.77 × 10^0^	5.16 × 10^−7^
	std	1.84 × 10^−1^	9.30 × 10^0^	2.99 × 10^0^	4.01 × 10^−7^
	min	3.01 × 10^−7^	5.04 × 10^1^	5.27 × 10^−1^	8.20 × 10^−8^
	max	8.20 × 10^−1^	9.48 × 10^1^	1.05 × 10^1^	1.50 × 10^−6^
F5	mean	4.86 × 10^1^	1.75 × 10^4^	3.93 × 10^2^	4.70 × 10^1^
	std	3.42 × 10^−1^	1.18 × 10^4^	4.15 × 10^2^	6.79 × 10^−1^
	min	4.78 × 10^1^	2.66 × 10^3^	1.92 × 10^2^	4.58 × 10^1^
	max	4.89 × 10^1^	4.68 × 10^4^	2.39 × 10^3^	4.86 × 10^1^
F6	mean	9.62 × 10^0^	7.36 × 10^1^	1.53 × 10^0^	2.05 × 10^0^
	std	2.75 × 10^−1^	3.16 × 10^1^	1.83 × 10^−1^	4.53 × 10^−1^
	min	8.96 × 10^0^	2.28 × 10^1^	1.17 × 10^0^	1.18 × 10^0^
	max	1.01 × 10^1^	1.44 × 10^2^	1.86 × 10^0^	2.76 × 10^0^
F7	mean	2.81 × 10^−3^	2.47 × 10^−1^	3.11 × 10^0^	2.61 × 10^−3^
	std	2.41 × 10^−3^	5.41 × 10^−2^	8.20 × 10^−1^	1.89 × 10^−3^
	min	1.59 × 10^−4^	1.21 × 10^−1^	1.88 × 10^0^	3.74 × 10^−4^
	max	1.11 × 10^−2^	3.40 × 10^−1^	4.98 × 10^0^	9.15 × 10^−3^
F8	mean	−3.77 × 10^3^	−7.67 × 10^3^	−6.65 × 10^3^	−9.02 × 10^3^
	std	4.48 × 10^2^	6.35 × 10^2^	3.09 × 10^3^	1.25 × 10^3^
	min	−5.46 × 10^3^	−9.57 × 10^3^	−1.25 × 10^4^	−1.09 × 10^4^
	max	−3.25 × 10^3^	−6.58 × 10^3^	−2.90 × 10^3^	−6.20 × 10^3^
F9	mean	5.52 × 10^−7^	4.32 × 10^2^	4.82 × 10^2^	0.00 × 10^0^
	std	2.56 × 10^−6^	1.33 × 10^1^	4.13 × 10^1^	0.00 × 10^0^
	min	6.39 × 10^−14^	3.93 × 10^2^	3.92 × 10^2^	0.00 × 10^0^
	max	1.43 × 10^−5^	4.57 × 10^2^	5.48 × 10^2^	0.00 × 10^0^
F10	mean	6.03 × 10^−5^	1.17 × 10^1^	1.99 × 10^1^	2.69 × 10^−14^
	std	2.68 × 10^−4^	8.32 × 10^0^	1.48 × 10^−1^	5.08 × 10^−15^
	min	5.93 × 10^−9^	2.48 × 10^0^	1.95 × 10^1^	1.42 × 10^−14^
	max	1.50 × 10^−3^	2.00 × 10^1^	2.02 × 10^1^	3.91 × 10^−14^
F11	mean	1.26 × 10^−6^	1.63 × 10^0^	7.43 × 10^−2^	0.00 × 10^0^
	std	3.48 × 10^−6^	2.85 × 10^−1^	1.16 × 10^−2^	0.00 × 10^0^
	min	0.00 × 10^0^	1.26 × 10^0^	5.44 × 10^−2^	0.00 × 10^0^
	max	1.40 × 10^−5^	2.64 × 10^0^	1.02 × 10^−1^	0.00 × 10^0^
F12	mean	4.96 × 10^−9^	3.43 × 10^1^	3.57 × 10^1^	0.00 × 10^0^
	std	1.19 × 10^−8^	4.35 × 10^0^	2.15 × 10^0^	0.00 × 10^0^
	min	0.00 × 10^0^	2.53 × 10^1^	3.16 × 10^1^	0.00 × 10^0^
	max	5.42 × 10^−8^	4.26 × 10^1^	3.99 × 10^1^	0.00 × 10^0^
F13	mean	3.43 × 10^−6^	2.33 × 10^1^	3.10 × 10^1^	2.67 × 10^−15^
	std	1.23 × 10^−5^	6.15 × 10^0^	3.39 × 10^0^	1.38 × 10^−14^
	min	1.42 × 10^−11^	1.34 × 10^1^	2.47 × 10^1^	6.62 × 10^−20^
	max	6.55 × 10^−5^	3.63 × 10^1^	4.05 × 10^1^	7.72 × 10^−14^
F14	mean	4.50 × 10^−7^	1.41 × 10^2^	4.49 × 10^1^	9.82 × 10^−6^
	std	1.70 × 10^−6^	7.07 × 10^1^	1.10 × 10^1^	2.50 × 10^−5^
	min	4.98 × 10^−16^	3.43 × 10^1^	3.05 × 10^1^	1.06 × 10^−7^
	max	9.18 × 10^−6^	3.35 × 10^2^	7.79 × 10^1^	1.42 × 10^−4^
F15	mean	3.99 × 10^−16^	3.16 × 10^−18^	2.48 × 10^−20^	2.24 × 10^−23^
	std	2.58 × 10^−16^	3.12 × 10^−18^	4.57 × 10^−20^	6.01 × 10^−23^
	min	3.36 × 10^−17^	2.09 × 10^−19^	1.85 × 10^−21^	2.24 × 10^−24^
	max	1.24 × 10^−15^	1.49 × 10^−17^	2.37 × 10^−19^	3.26 × 10^−22^

**Table 5 biomimetics-08-00084-t005:** Main parameters of the five algorithms.

Algorithm	The Main Parameters
GWO	$\vec{a}$ linearly decreases from 2 to 0
OGWO	$\vec{a}$ nonlinearly decreases from 2 to 0, u = 2
DGWO1	$\vec{a}$ linearly decreases from 2 to 0
DGWO2	$\vec{a}$ linearly decreases from 2 to 0
pGWO-CSA	$\vec{a}$ nonlinearly decreases from 2 to 0, u = 2

**Table 6 biomimetics-08-00084-t006:** The experimental results under 30 dimensions.

Function	Index	GWO	OGWO	DGWO1	DGWO2	pGWO-CSA
F1	mean	1.03 × 10^−27^	2.99 × 10^−40^	8.07 × 10^−21^	2.49 × 10^−62^	5.97 × 10^−44^
	std	1.25 × 10^−27^	7.79 × 10^−40^	1.14 × 10^−20^	5.68 × 10^−62^	1.12 × 10^−43^
	min	2.63 × 10^−29^	1.25 × 10^−45^	3.28 × 10^−22^	1.61 × 10^−65^	6.65 × 10^−47^
	max	5.66 × 10^−27^	3.96 × 10^−39^	5.94 × 10^−20^	2.84 × 10^−61^	4.87 × 10^−43^
F2	mean	8.66 × 10^−17^	3.38 × 10^−23^	1.16 × 10^−12^	2.48 × 10^−35^	3.66 × 10^−27^
	std	5.90 × 10^−17^	6.77 × 10^−23^	7.36 × 10^−13^	3.84 × 10^−35^	3.51 × 10^−27^
	min	2.39 × 10^−17^	6.53 × 10^−27^	1.72 × 10^−13^	1.59 × 10^−36^	3.98 × 10^−28^
	max	2.45 × 10^−16^	2.95 × 10^−22^	3.06 × 10^−12^	2.03 × 10^−34^	1.54 × 10^−26^
F3	mean	1.75 × 10^−26^	2.08 × 10^−38^	6.58 × 10^−20^	4.60 × 10^−60^	2.24 × 10^−42^
	std	2.61 × 10^−26^	7.19 × 10^−38^	4.32 × 10^−20^	2.36 × 10^−59^	8.92 × 10^−42^
	min	4.37 × 10^−28^	6.61 × 10^−45^	8.41 × 10^−21^	6.87 × 10^−65^	3.59 × 10^−45^
	max	1.33 × 10^−25^	3.87 × 10^−37^	1.73 × 10^−19^	1.32 × 10^−58^	5.00 × 10^−41^
F4	mean	1.21 × 10^−06^	1.44 × 10^−11^	3.02 × 10^−5^	2.76 × 10^−14^	1.08 × 10^−11^
	std	1.47 × 10^−6^	3.57 × 10^−11^	1.83 × 10^−5^	6.15 × 10^−14^	1.21 × 10^−11^
	min	9.64 × 10^−8^	6.57 × 10^−16^	5.36 × 10^−6^	1.27 × 10^−16^	1.00 × 10^−12^
	max	6.12 × 10^−6^	1.64 × 10^−10^	7.48 × 10^−5^	3.19 × 10^−13^	4.94 × 10^−11^
F5	mean	2.72 × 10^1^	2.69 × 10^1^	2.68 × 10^1^	2.70 × 10^1^	2.67 × 10^1^
	std	6.10 × 10^−1^	5.60 × 10^−1^	7.98 × 10^−1^	7.49 × 10^−1^	5.18 × 10^−1^
	min	2.60 × 10^1^	2.62 × 10^1^	2.59 × 10^1^	2.61 × 10^1^	2.59 × 10^1^
	max	2.87 × 10^1^	2.80 × 10^1^	2.86 × 10^1^	2.85 × 10^1^	2.80 × 10^1^
F6	mean	7.83 × 10^−1^	6.53 × 10^−1^	5.27 × 10^−1^	5.98 × 10^−1^	4.20 × 10^−1^
	std	4.10 × 10^−1^	3.43 × 10^−1^	3.30 × 10^−1^	2.59 × 10^−1^	3.28 × 10^−1^
	min	8.81 × 10^−3^	4.32 × 10^−4^	6.32 × 10^−5^	2.46 × 10^−1^	3.14 × 10^−6^
	max	1.66 × 10^0^	1.49 × 10^0^	1.26 × 10^0^	1.00 × 10^0^	1.50 × 10^0^
F7	mean	1.64 × 10^−3^	1.67 × 10^−4^	2.06 × 10^−3^	1.60 × 10^−3^	1.38 × 10^−3^
	std	6.39 × 10^−4^	1.38 × 10^−4^	8.83 × 10^−4^	7.95 × 10^−4^	8.15 × 10^−4^
	min	6.23 × 10^−4^	1.25 × 10^−5^	5.40 × 10^−4^	4.23 × 10^−4^	3.62 × 10^−4^
	max	3.20 × 10^−3^	4.88 × 10^−4^	4.67 × 10^−3^	3.83 × 10^−3^	3.67 × 10^−3^
F8	mean	−5.71 × 10^3^	−4.10 × 10^3^	−5.85 × 10^3^	−5.74 × 10^3^	−6.13 × 10^3^
	std	9.10 × 10^2^	1.38 × 10^3^	8.53 × 10^2^	8.88 × 10^2^	7.70 × 10^2^
	min	−6.73 × 10^3^	−7.63 × 10^3^	−7.37 × 10^3^	−6.84 × 10^3^	−7.55 × 10^3^
	max	−3.42 × 10^3^	−2.82 × 10^3^	−3.60 × 10^3^	−3.15 × 10^3^	−4.68 × 10^3^
F9	mean	3.51 × 10^0^	1.44 × 10^−1^	7.76 × 10^0^	4.31 × 10^−1^	0.00 × 10^0^
	std	8.96 × 10^0^	7.75 × 10^−1^	2.26 × 10^1^	1.50 × 10^0^	0.00 × 10^0^
	min	0.00 × 10^0^	0.00 × 10^0^	3.73 × 10^−14^	0.00 × 10^0^	0.00 × 10^0^
	max	4.67 × 10^1^	4.32 × 10^0^	1.28 × 10^2^	6.51 × 10^0^	0.00 × 10^0^
F10	mean	9.90 × 10^−14^	8.64 × 10^−15^	1.55 × 10^−11^	2.45 × 10^−14^	8.17 × 10^−15^
	std	1.26 × 10^−14^	3.96 × 10^−15^	8.32 × 10^−12^	4.04 × 10^−15^	2.62 × 10^−15^
	min	6.75 × 10^−14^	3.55 × 10^−15^	4.68 × 10^−12^	1.42 × 10^−14^	3.55 × 10^−15^
	max	1.28 × 10^−13^	1.78 × 10^−14^	4.06 × 10^−11^	2.84 × 10^−14^	1.42 × 10^−14^
F11	mean	3.72 × 10^−3^	1.12 × 10^−3^	2.73 × 10^−3^	7.71 × 10^−3^	0.00 × 10^0^
	std	7.49 × 10^−3^	4.34 × 10^−3^	6.32 × 10^−3^	1.30 × 10^−2^	0.00 × 10^0^
	min	0.00 × 10^0^	0.00 × 10^0^	0.00 × 10^0^	0.00 × 10^0^	0.00 × 10^0^
	max	2.79 × 10^−2^	2.14 × 10^−2^	2.19 × 10^−2^	4.92 × 10^−2^	0.00 × 10^0^
F12	mean	0.00 × 10^0^	0.00 × 10^0^	0.00 × 10^0^	0.00 × 10^0^	0.00 × 10^0^
	std	0.00 × 10^0^	0.00 × 10^0^	0.00 × 10^0^	0.00 × 10^0^	0.00 × 10^0^
	min	0.00 × 10^0^	0.00 × 10^0^	0.00 × 10^0^	0.00 × 10^0^	0.00 × 10^0^
	max	0.00 × 10^0^	0.00 × 10^0^	0.00 × 10^0^	0.00 × 10^0^	0.00 × 10^0^
F13	mean	5.30 × 10^4^	5.69 × 10^−5^	1.07 × 10^−3^	3.09 × 10^−5^	1.28 × 10^−23^
	std	7.11 × 10^−4^	1.83 × 10^−4^	8.13 × 10^−4^	1.01 × 10^−4^	3.86 × 10^−23^
	min	2.66 × 10^−17^	1.79 × 10^−27^	6.06 × 10^−12^	3.18 × 10^−37^	1.70 × 10^−27^
	max	2.46 × 10^−3^	8.65 × 10^−4^	2.67 × 10^3^	5.46 × 10^−4^	1.68 × 10^−22^
F14	mean	2.00 × 10^−5^	9.68 × 10^−6^	2.69 × 10^−5^	3.23 × 10^−6^	5.36 × 10^−6^
	std	1.93 × 10^−5^	1.79 × 10^−5^	2.25 × 10^−5^	5.40 × 10^−6^	7.72 × 10^−6^
	min	7.40 × 10^−7^	1.79 × 10^−7^	4.62 × 10^−6^	4.23 × 10^−8^	2.86 × 10^−8^
	max	7.44 × 10^−5^	9.70 × 10^−5^	1.01 × 10^−4^	2.79 × 10^−5^	2.89 × 10^−5^
F15	mean	9.80 × 10^−16^	9.12 × 10^−12^	9.65 × 10^−16^	2.82 × 10^−15^	1.75 × 10^−16^
	std	4.96 × 10^−16^	1.64 × 10^−11^	4.40 × 10^−16^	1.07 × 10^−14^	5.99 × 10^−16^
	min	4.87 × 10^−16^	3.76 × 10^−15^	4.41 × 10^−16^	3.12 × 10^−16^	2.30 × 10^−17^
	max	3.11 × 10^−15^	6.83 × 10^−11^	2.97 × 10^−15^	6.05 × 10^−14^	3.39 × 10^−15^

**Table 7 biomimetics-08-00084-t007:** The experimental results under 50 dimensions.

Function	Index	GWO	OGWO	DGWO1	DGWO2	pGWO-CSA
F1	mean	7.84 × 10^−20^	1.19 × 10^−29^	1.54 × 10^−14^	8.58 × 10^−47^	3.29 × 10^−31^
	std	8.07 × 10^−20^	4.26 × 10^−29^	1.32 × 10^−14^	1.98 × 10^−46^	3.21 × 10^−31^
	min	8.05 × 10^−21^	1.15 × 10^−34^	3.03 × 10^−15^	5.11 × 10^−49^	2.68 × 10^−33^
	max	3.11 × 10^−19^	2.13 × 10^−28^	7.68 × 10^−14^	8.72 × 10^−46^	1.33 × 10^−30^
F2	mean	2.20 × 10^−12^	1.15 × 10^−17^	5.02 × 10^−9^	8.20 × 10^−27^	7.23 × 10^−20^
	std	1.30 × 10^−12^	2.79 × 10^−17^	2.85 × 10^−9^	9.32 × 10^−27^	4.78 × 10^−20^
	min	4.72 × 10^−13^	2.66 × 10^−19^	1.57 × 10^−9^	7.22 × 10^−28^	9.01 × 10^−21^
	max	6.73 × 10^−12^	1.50 × 10^−16^	1.58 × 10^−8^	4.41 × 10^−26^	1.95 × 10^−19^
F3	mean	1.22 × 10^−18^	3.52 × 10^−28^	5.31 × 10^−13^	1.93 × 10^−45^	6.91 × 10^−30^
	std	1.37 × 10^−18^	1.16 × 10^−27^	8.55 × 10^−13^	3.81 × 10^−45^	7.60 × 10^−30^
	min	1.07 × 10^−19^	1.30 × 10^−36^	3.51 × 10^−14^	3.01 × 10^−48^	1.26 × 10^−31^
	max	6.30 × 10^−18^	6.16 × 10^−27^	4.24 × 10^−12^	1.98 × 10^−44^	2.58 × 10^−29^
F4	mean	4.49 × 10^−4^	3.67 × 10^−8^	5.11 × 10^−3^	1.48 × 10^−9^	5.16 × 10^−7^
	std	3.16 × 10^−4^	9.79 × 10^−8^	5.40 × 10^−3^	2.37 × 10^−9^	4.01 × 10^−7^
	min	7.15 × 10^−5^	2.84 × 10^−12^	1.55 × 10^−3^	1.79 × 10^−11^	8.20 × 10^−8^
	max	1.33 × 10^−3^	4.01 × 10^−7^	2.93 × 10^−2^	1.07 × 10^−8^	1.50 × 10^−6^
F5	mean	4.75 × 10^1^	4.71 × 10^1^	4.71 × 10^1^	4.71 × 10^1^	4.70 × 10^1^
	std	8.65 × 10^−1^	6.93 × 10^−1^	7.38 × 10^−1^	6.92 × 10^−1^	6.79 × 10^−1^
	min	4.60 × 10^1^	4.61 × 10^1^	4.61 × 10^1^	4.61 × 10^1^	4.58 × 10^1^
	max	4.87 × 10^1^	4.85 × 10^1^	4.86 × 10^1^	4.85 × 10^1^	4.86 × 10^1^
F6	mean	2.66 × 10^0^	2.28 × 10^0^	2.05 × 10^0^	2.71 × 10^0^	2.05 × 10^0^
	std	5.41 × 10^−1^	5.24 × 10^−1^	6.12 × 10^−1^	4.58 × 10^−1^	4.53 × 10^−1^
	min	1.25 × 10^0^	1.38 × 10^0^	6.89 × 10^−1^	1.75 × 10^0^	1.18 × 10^0^
	max	4.00 × 10^0^	3.50 × 10^0^	3.25 × 10^0^	3.95 × 10^0^	2.76 × 10^0^
F7	mean	3.52 × 10^−3^	2.27 × 10^−4^	4.75 × 10^−3^	2.75 × 10^−3^	2.61 × 10^−3^
	std	1.80 × 10^−3^	2.70 × 10^−4^	2.20 × 10^−3^	1.03 × 10^−3^	1.89 × 10^−3^
	min	8.32 × 10^−4^	1.65 × 10^−5^	1.64 × 10^−3^	1.02 × 10^−3^	3.74 × 10^−4^
	max	8.12 × 10^−3^	1.42 × 10^−3^	1.21 × 10^−2^	4.90 × 10^−3^	9.15 × 10^−3^
F8	mean	−8.79 × 10^3^	−5.64 × 10^3^	−8.46 × 10^3^	−8.91 × 10^3^	−9.21 × 10^3^
	std	1.49 × 10^3^	2.17 × 10^3^	1.96 × 10^3^	9.69 × 10^2^	9.61 × 10^2^
	min	−1.09 × 10^4^	−1.05 × 10^4^	−1.08 × 10^4^	−1.10 × 10^4^	−1.13 × 10^4^
	max	−4.19 × 10^3^	−3.95 × 10^3^	−3.82 × 10^3^	−7.33 × 10^3^	−7.62 × 10^3^
F9	mean	4.54 × 10^0^	4.71 × 10^−5^	9.85 × 10^0^	9.88 × 10^−1^	0.00 × 10^0^
	std	5.08 × 10^0^	2.54 × 10^−4^	6.01 × 10^0^	2.86 × 10^0^	0.00 × 10^0^
	min	2.49 × 10^−14^	0.00 × 10^0^	8.42 × 10^−11^	0.00 × 10^0^	0.00 × 10^0^
	max	1.94 × 10^1^	1.41 × 10^−3^	2.57 × 10^1^	1.38 × 10^1^	0.00 × 10^0^
F10	mean	3.77 × 10^−11^	2.14 × 10^−14^	1.55 × 10^−8^	3.94 × 10^−14^	2.69 × 10^−14^
	std	2.50 × 10^−11^	1.72 × 10^−14^	5.43 × 10^−9^	2.80 × 10^−15^	5.08 × 10^−15^
	min	1.18 × 10^−11^	3.55 × 10^−15^	5.64 × 10^−9^	3.20 × 10^−14^	1.42 × 10^−14^
	max	1.31 × 10^−10^	6.75 × 10^−14^	2.87 × 10^−8^	4.26 × 10^−14^	3.91 × 10^−14^
F11	mean	3.56 × 10^−3^	2.37 × 10^−3^	3.47 × 10^−3^	2.83 × 10^−3^	0.00 × 10^0^
	std	7.35 × 10^−3^	7.32 × 10^−3^	8.44 × 10^−3^	6.50 × 10^−3^	0.00 × 10^0^
	min	0.00 × 10^0^	0.00 × 10^0^	6.33 × 10^−15^	0.00 × 10^0^	0.00 × 10^0^
	max	2.31 × 10^−2^	3.12 × 10^−2^	3.58 × 10^−2^	2.15 × 10^−2^	0.00 × 10^0^
F12	mean	0.00 × 10^0^	0.00 × 10^0^	1.75 × 10^−14^	0.00 × 10^0^	0.00 × 10^0^
	std	0.00 × 10^0^	0.00 × 10^0^	1.31 × 10^−14^	0.00 × 10^0^	0.00 × 10^0^
	min	0.00 × 10^0^	0.00 × 10^0^	4.77 × 10^−15^	0.00 × 10^0^	0.00 × 10^0^
	max	0.00 × 10^0^	0.00 × 10^0^	5.51 × 10^−14^	0.00 × 10^0^	0.00 × 10^0^
F13	mean	7.99 × 10^−4^	2.10 × 10^−4^	2.00 × 10^−3^	1.00 × 10^−4^	2.67 × 10^−15^
	std	9.17 × 10^−4^	4.57 × 10^−4^	1.23 × 10^−3^	2.39 × 10^−4^	1.38 × 10^−14^
	min	1.83 × 10^−12^	7.86 × 10^−22^	2.17 × 10^−9^	6.97 × 10^−28^	6.62 × 10^−20^
	max	3.35 × 10^−3^	1.94 × 10^−3^	4.81 × 10^−3^	1.02 × 10^−3^	7.72 × 10^−14^
F14	mean	3.55 × 10^−5^	1.37 × 10^−5^	6.99 × 10^−5^	5.78 × 10^−6^	9.82 × 10^−6^
	std	3.21 × 10^−5^	1.09 × 10^−5^	4.69 × 10^−5^	6.70 × 10^−6^	2.50 × 10^−5^
	min	1.46 × 10^−6^	4.20 × 10^−7^	9.40 × 10^−6^	4.63 × 10^−8^	1.06 × 10^−7^
	max	1.24 × 10^−4^	3.78 × 10^−5^	2.15 × 10^−4^	2.39 × 10^−5^	1.42 × 10^−4^
F15	mean	1.14 × 10^−22^	9.50 × 10^−18^	5.62 × 10^−23^	5.52 × 10^−23^	2.24 × 10^−23^
	std	2.82 × 10^−22^	2.07 × 10^−17^	6.32 × 10^−23^	5.49 × 10^−23^	6.01 × 10^−23^
	min	2.08 × 10^−23^	1.82 × 10^−21^	1.80 × 10^−23^	1.83 × 10^−23^	2.24 × 10^−24^
	max	1.60 × 10^−21^	8.02 × 10^−17^	3.20 × 10^−22^	2.82 × 10^−22^	3.26 × 10^−22^

**Table 8 biomimetics-08-00084-t008:** The results of the Wilcoxon test.

Function	Dimension	PSO	DE	FA	GWO	OGWO	DGWO1	DGWO2
F1	30	−	−	−	−	−	−	+
	50	−	−	−	−	−	−	+
F2	30	−	−	−	−	−	−	+
	50	−	−	−	−	−	−	+
F3	30	−	−	−	−	−	−	+
	50	−	−	−	−	−	−	+
F4	30	−	−	−	−	−	−	+
	50	−	−	−	−	+	−	+
F5	30	−	−	−	−	−	−	−
	50	−	−	−	−	−	−	−
F6	30	−	+	−	−	−	−	−
	50	−	−	+	−	−	=	−
F7	30	−	−	−	−	+	−	−
	50	−	−	−	−	+	−	−
F8	30	−	−	−	−	−	−	−
	50	−	−	−	−	−	−	−
F9	30	−	−	−	−	−	−	−
	50	−	−	−	−	−	−	−
F10	30	−	−	−	−	−	−	−
	50	−	−	−	−	+	−	−
F11	30	−	−	−	−	−	−	−
	50	−	−	−	−	−	−	−
F12	30	−	−	−	=	=	=	=
	50	−	−	−	=	=	−	=
F13	30	−	−	−	−	−	−	−
	50	−	−	−	−	−	−	−
F14	30	+	−	−	−	−	−	+
	50	+	−	−	−	−	−	+
F15	30	−	−	−	−	−	−	−
	50	−	−	−	−	−	−	−
+		2	1	1	0	4	0	10
−		28	29	29	28	24	28	18
=		0	0	0	2	2	2	2

**Table 9 biomimetics-08-00084-t009:** Experimental results.

	PSO	DE	FA	GWO	OGWO	DGWO1	DGWO2	pGWO-CSA
min	8.81 × 10^0^	7.56 × 10^0^	7.17 × 10^0^	7.57 × 10^0^	7.57 × 10^0^	7.56 × 10^0^	8.36 × 10^0^	7.16 × 10^0^
max	1.03 × 10^1^	9.44 × 10^0^	1.34 × 10^1^	9.27 × 10^0^	8.74 × 10^0^	9.22 × 10^0^	1.01 × 10^1^	8.51 × 10^0^
mean	9.97 × 10^0^	8.46 × 10^0^	1.08 × 10^1^	8.30 × 10^0^	8.10 × 10^0^	8.13 × 10^0^	9.51 × 10^0^	7.94 × 10^0^

## Data Availability

Not applicable.

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
