# Peer review of "An Improved Grey Wolf Optimizer and Its Application in Robot Path Planning"

_biomimetics, 2023, doi:10.3390/biomimetics8010084_

Round 1

Reviewer 1 Report

This paper discusses a hybrid grey wolf optimizer utilizing clone selection algorithm (pGWO-CSA) to overcome the disadvantages of standard grey wolf optimizer (GWO). In order to verify the applicability of the algorithm, pGWO-CSA is applied to the robot path-planning problem and obtain excellent results. I think this topic is beyond the scope of the journal Biomimetics, and this work is probably more suitable for specific journals. Therefore, I do not recommend it to be considered for publication in Bimimetics.

Author Response

Dear Editor and Reviewer

      The authors would like to appreciate the editors again for giving us this opportunity to revise the paper, and also thank the reviewers for their careful review and recognition of our manuscript, and for providing us with comments and suggestions to improve the quality of the manuscript. Here, we submit a correction to the manuscript entitled "An improved grey wolf optimizer and its application in robot path planning (biomimetics-2113666)", which has been revised as suggested by the reviewers. Please see the attachment. Thank you!

Reviewer 2 Report

The paper presents an improved algorithm of grey wolf optimizer, which obtains better results compared to other swarm intelligence algorithms, GWO and its variants. The proposed pGWO-CSA replaced linear function with nonlinear function and combined GWO with CSA to improve the convergence speed and accuracy in the single-peak function and the ability to jump out of local optimum in the multi-peak function and complex problem, which is a novel idea. However, a few revisions are needed as below.

1.  In section 3.1.1, the iterative attenuation of parameter is calculated by formula (9), can author briefly introduce the source or basis of this formula?

2. How to determine the maximum number of iterations and what effect the number of iterations has on the final result?

3.  There is a problem with the subtitles of Figure 5-Figure 34, please correct them.

4.     I noticed that author used a similar research method as the literature of “An Improved Grey Wolf Optimization Algorithm and Its Application in Path Planning”. Could author describe what is the difference? That literature utilized simple and complex environment to evaluate algorithm in his path planning experiment, so can author provide more related experiments with different environment to test the present algorithm?

Author Response

(The authors gave the same response as above.)

Reviewer 3 Report

The authors have proposed a study entitled " An improved grey wolf optimizer and its application in robot path planning". there are some minor typos and English writing problems in the manuscript which should be considered by the authors for publication. The authors should also consider following comments for their manuscript.

Q1. the form of parameters description should be more consistent. Some of them should be improved.

Q2. some language defects should be improved;

Q3. the conclusion should be improved.

Q4. the beginning of the introduction point should be presented in the form of some not cited, free explanations about the grounds, source of the problem with also some practical meaning.

Q5. The motivation behind using multiple metaheuristic algorithms to find the best combination of parameters of the algorithm. The number of decision variables is considered low and there is no constraints. 

Q6. The motivation on choosing GWO over many prediction methods, such as ANN, SVM, RF, LS-SVM, is unclear.

Q7. what is the objective function of the training process?

Q8. it is unclear if the values are used for testing dataset, validation dataset, or the average of k-fold dataset.

Q9. The iteration and population of each algorithm is different. How can we ensure that the performance comparison of each algorithm is fairly conducted?

Q10. All symbols and parameters in equations should be defined (it is necessary).

Author Response

(The authors gave the same response as above.)

Round 2

Reviewer 1 Report

It can be accepted for publication.

Reviewer 2 Report

The authors had made a revision based on the requirement from the first review. However, there are still some serious problems needed to be addressed.

1. The writing is hard to understand and lack of the organization, also there are lots of grammer mistake made by authors. I strongly suggest that the authors find a native english speaker to re-edit the paper and correct all grammer errors. For example, there are lots of "the" missing in the paragraph. Also, the experimental test parts are redundent and very chaotic. I suggest authors to reorganize this part.

2. For the first review,  I required more experitmental results for the path planning part. However, this part is still missing and authors didn't respond properly. The path planning is very fundamental problem for robotics and intensely used to test the algorithms or optimizers. The environment used in the manuscript is the too simple and can't show the significance from the point of view of algorithm searching efficiciency. I suggest that authors add more tests with complex environment and also compare with some classic path planning algorithm like A*, Djikstra. If there is no obvious improvement on the path planning issue, authros should give more analysis and explanations.